# Y RNA fragment in extracellular vesicles confers cardioprotection via modulation of IL-10 expression and secretion

Linda Cambier[1,†], Geoffrey de Couto[1,†], Ahmed Ibrahim[2], Antonio K Echavez[1], Jackelyn Valle[1], Weixin Liu[1], Michelle Kreke[2], Rachel R Smith[2], Linda Marbán[2] & Eduardo Marbán[1,*]

## Abstract

Cardiosphere-derived cells (CDCs) reduce myocardial infarct size via secreted extracellular vesicles (CDC-EVs), including exosomes, which alter macrophage polarization. We questioned whether short non-coding RNA species of unknown function within CDC-EVs contribute to cardioprotection. The most abundant RNA species in CDC-EVs is a Y RNA fragment (EV-YF1); its relative abundance in CDC-EVs correlates with CDC potency *in vivo*. Fluorescently labeled EV-YF1 is actively transferred from CDCs to target macrophages via CDC-EVs. Direct transfection of macrophages with EV-YF1 induced transcription and secretion of IL-10. When cocultured with rat cardiomyocytes, EV-YF1-primed macrophages were potently cytoprotective toward oxidatively stressed cardiomyocytes through induction of IL-10. *In vivo*, intracoronary injection of EV-YF1 following ischemia/reperfusion reduced infarct size. A fragment of Y RNA, highly enriched in CDC-EVs, alters *Il10* gene expression and enhances IL-10 protein secretion. The demonstration that EV-YF1 confers cardioprotection highlights the potential importance of diverse exosomal contents of unknown function, above and beyond the usual suspects (e.g., microRNAs and proteins).

**Keywords** extracellular vesicle; macrophage; RNA; stem cells

**Subject Categories** Cardiovascular System; Stem Cells

## Introduction

Myocardial infarction (MI) affects > 7 million people in the United States alone (Mozaffarian *et al*, 2015). Cardiosphere-derived cells (CDCs) (Smith *et al*, 2007; Kreke *et al*, 2012) show promise in mitigating MI damage: They can regenerate "irreversibly" injured myocardium by increasing viable tissue and decreasing scar size (Makkar *et al*, 2012; Malliaras *et al*, 2012, 2013b; Tseliou *et al*, 2013, 2014). The benefits are not limited to chronic injury. In both porcine and rat models of acute ischemia–reperfusion (I/R) injury (de Couto *et al*, 2015; Kanazawa *et al*, 2015), CDCs modulate the inflammatory response and confer cardioprotection.

CDCs act by secreting lipid-bilayer vesicles known as extracellular vesicles (CDC-EVs) (Ibrahim *et al*, 2014). Exosomes are a specific subpopulation of small (~30–150 nm) EVs which are products of the endolysosomal pathway (Ibrahim & Marbán, 2016). These naturally occurring EVs harbor and transfer a diverse cargo of non-coding RNAs (including miRNAs) and proteins (Valadi *et al*, 2007), highlighting new means of cellular crosstalk with major implications for clinical therapeutics (Sahoo *et al*, 2011; Sahoo & Losordo, 2014; Emanueli *et al*, 2015; Khan *et al*, 2015; Ibrahim & Marbán, 2016; Kishore & Khan, 2016). We recently demonstrated that CDC-EVs are anti-fibrotic and angiogenic (Tseliou *et al*, 2015) and promote cardiomyocyte survival and proliferation following MI (Ibrahim *et al*, 2014), while identifying miR-146a as a highly enriched miRNA within CDC-EVs that promotes cardiac regeneration *in vivo* (Ibrahim *et al*, 2014). However, miRNAs comprise only a small portion of the RNA within exosomes and other EVs (Cheng *et al*, 2014b).

One poorly understood class of non-coding RNA, the Y RNAs, is particularly plentiful in EVs (Valadi *et al*, 2007). These small (83–112 nucleotides) RNAs were first discovered in complex with ribonucleoproteins in the serum of patients suffering from lupus or Sjögren's syndrome (Hendrick *et al*, 1981; Lerner *et al*, 1981). Four distinct human Y RNAs share a characteristic stem-loop secondary structure and high sequence conservation between the upper and lower stem (Teunissen *et al*, 2000). In addition, the human genome contains > 1,000 Y pseudogenes (Jurka *et al*, 1988; O'Brien & Harley, 1992; Perreault *et al*, 2005). Here, we describe a Y RNA fragment (EV-YF1) highly enriched in CDC-EVs and show that this oligoribonucleotide affects gene expression to induce cytoprotection.

1 Cedars-Sinai Heart Institute, Cedars-Sinai Medical Center, Los Angeles, CA, USA
2 Capricor Inc., Los Angeles, CA, USA
*Corresponding author. Tel: +1 310 423 7557; Fax: +1 310 423 7637; E-mail: eduardo.marban@cshs.org
†These authors contributed equally to this work

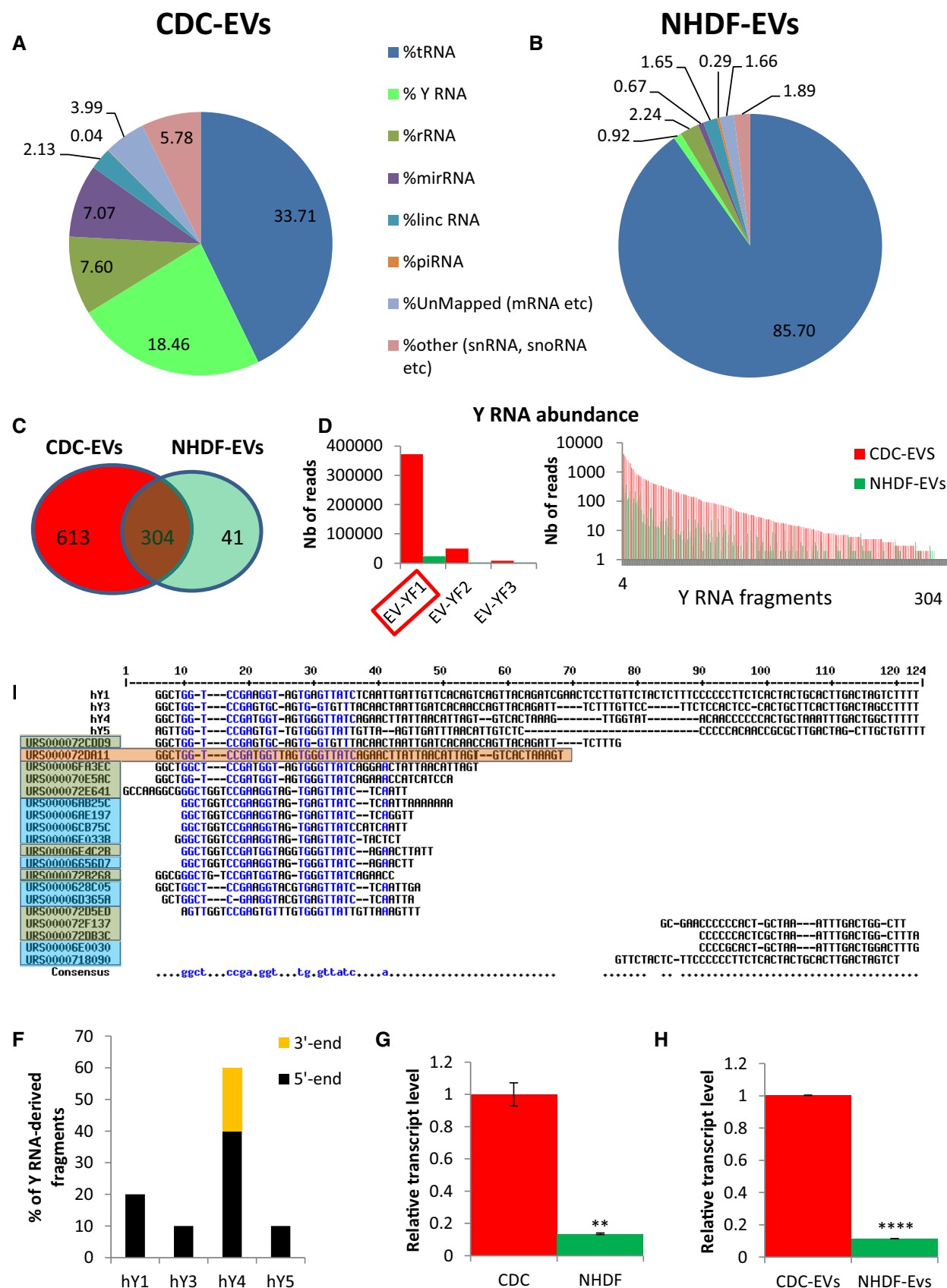

Figure 1.

◀

**Figure 1. RNA content of CDC-EVs (day 5).**

A, B Pie chart depicting the percent distribution of small RNA species in CDC-EVs (A) and NHDF-EVs (B), collected following 5 days of serum-free culture.

C Venn diagram depicting the number of unique and common Y RNA sequences in CDC-EVs and NHDF-EVs.

D Graphical depictions of the abundance of the common Y RNA fragments in CDC-EVs and NHDF-EVs according to the number of reads obtained by RNA-seq. Left graph: top three most abundant Y RNA fragments (linear scale). Right graph: the remaining 301 Y RNA fragments (logarithmic scale).

E Sequence alignment of each full-length human Y RNA (hY1, hY3, hY4, and hY5) with Y RNA fragments. Highlighted here are the top nine Y RNA fragments uniquely expressed in CDC-EVs (blue; 9/613 in C) and the top 10 commonly expressed between CDC-EVs and NHDF-EVs (green; 10/304 in C). The most highly expressed Y RNA fragment (EV-YF1) is highlighted in orange.

F Proportion of Y RNA fragments derived from the 5′- or 3′-end of the four full-length human Y RNA genes.

G, H Relative expression of EV-YF1 by qPCR in CDCs and NHDFs (G) and their respective EVs (H). Results depict the mean $\pm$ SEM of two independent experiments, $n$ = 6. Groups were compared using two-tailed, unpaired, Student's *t*-test; (G) **$P$ = 0.0024; (H) ****$P$ < 0.0001.

## Results

### Y RNA fragments are enriched in CDC-EVs

Exosome-enriched EVs from six human CDC donors exhibited typical particle numbers and size distributions compared to normal human dermal fibroblast (NHDF) EVs (NHDF-EVs), as exemplified in Appendix Fig S1A and B. RNA sequencing (RNA-seq) of species up to 200 nucleotides in length revealed that CDC-EVs contain many small RNAs: Fig 1A shows a representative pie chart from one donor (OD220), and Fig 2A shows pooled data from six different CDC donors with distinct demographic properties (Table 1) but identical surface marker expression (Table 2). For comparison, Fig 1B shows the ncRNA distribution in NHDF-EVs (Ibrahim *et al*, 2014). EVs from the two cell types differ markedly in their RNA profiles, with a much greater dominance of tRNA in NHDF-EVs. The most abundant RNA species in CDC-EVs after tRNA is Y RNA (~20% of total RNA). Indeed, Y RNAs are much more plentiful than miRNAs, which represent only ~5% of the total RNA (Figs 1A and 2A).

The abundance of Y RNAs in CDC-EVs motivated us to determine whether they play a role in mediating the effects of CDCs and CDC-EVs. RNA-seq revealed 917 Y RNA sequences in CDC-EVs and 345 in NHDF-EVs. The Y RNA sequences in both groups were fragments of Y RNA that varied in length (15–62 nt) (Fig 3A). Among those sequences, 613 were unique to CDC-EVs, 41 were unique to NHDF-EVs, and 304 were common to CDC-EVs and NHDF-EVs (Fig 1C); unique Y RNA species were, however, very low in abundance in both types of EVs (< 1,000-fold the number of reads as for the shared species; cf. Fig 3B). The Y RNA fragments present in both CDC-EVs and NHDF-EVs were generally more abundant in CDC-EVs (Fig 1D). As a case in point, the most plentiful Y RNA fragment in CDC-EVs (RNA central access: URS000072DA11; denoted herein as EV-YF1) was 15.7-fold more abundant in CDC-EVs than in NHDF-EVs (Fig 1D). Indeed, EV-YF1 is the *single most abundantly expressed ncRNA species* in CDC-EVs (Fig 2B).

Full-length human Y RNAs (hY) exhibit extensive sequence and structural conservation among members (van Gelder *et al*, 1994; Teunissen *et al*, 2000). Figure 1E shows BLAST sequence alignments of the four hY family members, the top 19 most abundant Y RNA fragments found only in CDC-EVs (Fig 1E, blue), and the top 10 most abundant Y RNA fragments found both in CDC-EVs and NHDF-EVs (Fig 1E, green). Sixteen of the 19 Y RNA fragments aligned to or near the 5′ end of the four hY family members (Fig 1E and F); however, there was a particular enrichment in those homologous to hY4 (Fig 1F). To validate these findings, we

**Table 1. Demographic properties of human CDC donors.**

| Donor | Age | Sex | Ethnicity | Cause of death |
|-------|-----|-----|-----------|----------------|
| YKT | 56 | M | Hispanic | Head trauma |
| BM030 | 27 | F | Caucasian | Anoxia |
| L088 | 64 | M | Caucasian | Stroke |
| ZCl | 9 | F | Chinese | Anoxia |
| ZKN | 26 | F | Hispanic | Head trauma/MVA/blunt injury |
| OD220 | 3 | M | Caucasian | MVA |

MVA, Motor vehicle accident.
Medical history was unremarkable in all donors except ZCl who had hydrocephalus due to craniometaphyseal dysplasia.

**Table 2. Phenotype of CDCs from different donors.**

| | CD105 | c-Kit | CD31 | CD90 | CD45 | DDR2 |
|------|-------|-------|------|------|------|------|
| OD220 | 99.70% | 2.90% | 1.08% | 25.30% | 0.66% | 6.67% |
| YKT | 99.80% | 3.69% | 1.28% | 17.60% | 0.52% | 3.92% |
| ZKN | 99.90% | 2.37% | 2.15% | 26.30% | 1.13% | 7.61% |
| L088 | 99.90% | 2.91% | 4.51% | 37.20% | 0.76% | 3.52% |
| ZCl | 99.70% | 1.80% | 0.87% | 10.90% | 0.79% | 3.17% |
| BM030 | 99.80% | 2.88% | 4.46% | 51.2% | 2.03% | 1.28% |

Percentage of marker abundance expressed in CDCs from different donors obtained by flow cytometry.

examined all the Y RNA fragments within CDC-EVs and NHDF-EVs and found that ~85% of all Y RNA fragments appeared to be derived from hY4 (Fig 3C). Based on these results, we opted to focus our attention on EV-YF1 because of its abundance. To confirm our RNA-seq data, we designed primers for EV-YF1 and analyzed its expression by qPCR in cells (CDCs and NHDFs; Fig 1G) and EVs (CDC-EVs and NHDF-EVs; Fig 1H). EV-YF1 expression was much greater in CDCs and CDC-EVs than in the respective NHDF controls (~10-fold, Fig 1G and H). The EV-YF1 fragment aligns well with the 5′ end of hY4 (98% homology, with the exception of an additional thymine [T] at position 16 in EV-YF1; Fig 3D). Thermodynamics-based UNAFold software (Markham & Zuker, 2008) yielded five energetically probable secondary structures for EV-YF1 (Fig 3E). While details of predicted structures differ, all share stem-loop motifs common in Y RNA species (Chen & Heard, 2013).

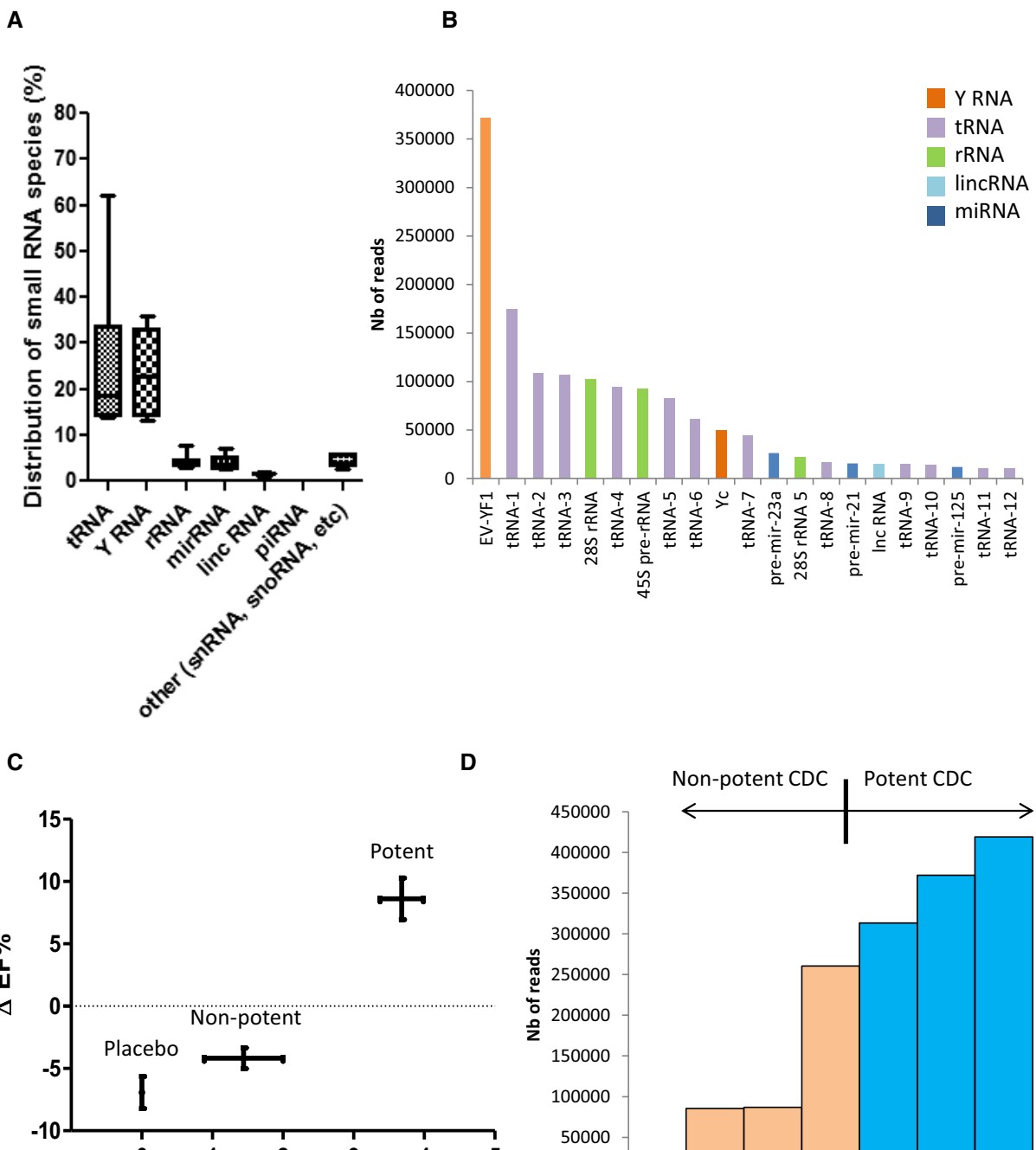

**Figure 2. CDC-EVs EV-YF1 content correlates with CDC potency *in vivo*.**

A  Graph depicting the percent distribution of small RNA species in CDC-EVs from six different donors (as shown in Table 1). Horizontal lines represent the median value, box limits represent 75 and 25% percentile of the total values and the bars represent the maximal and minimal value.

B  Graph representing the most abundant sequences expressed in OD220 CDC-EVs. EV-YF1: URS000072DA11; tRNA-1: URS00006FBEE8; tRNA-2: URS000072EF3B; tRNA-3: URS0000758E15; 28S rRNA: URS00003692B6; tRNA-4: URS000072CC66; 45S pre-rRNA: URS000025EB0F; tRNA-5: URS000072F18F; tRNA-6: URS000072F2C3; Yc: URS000072E641; tRNA-7: URS000072B56D; pre-mir-23a: URS000075EDA9; 28S rRNA 5: URS000075EC78; tRNA-8: URS0000701715; pre-mir-21: URS000075E5CC; long non-coding RNA (Mir17hg gene): URS000076343C; tRNA-9: URS00006A0CFD; tRNA-10: URS0000717173; pre-mir-12: URS00007A4AA9; tRNA-11: URS0000750232; tRNA-12: URS000072345A.

C  Correlation between the percent change in ejection fraction (baseline 2 h post-MI to 21 days, ΔEF%) post-MI with CDC treatment (six different donors, *n* = 8 animals/donor) or placebo (*n* = 14 animals) and EV-YF1 abundance in CDC-EVs. Potent CDCs (ZCI, YKT, OD220) were delineated from non-potent CDCs (LO88, BM030, ZKN) by positive ΔEF%. Error bars represent the SEM of the delta ejection fraction % between animals treated with placebo, non-potent or potent CDCs, respectively.

D  EV-YF1 abundance based on RNA-seq counts in EVs from potent (ZCI, YKT, OD220) and non-potent CDCs (LO88, BM030, ZKN) and NHDFs.

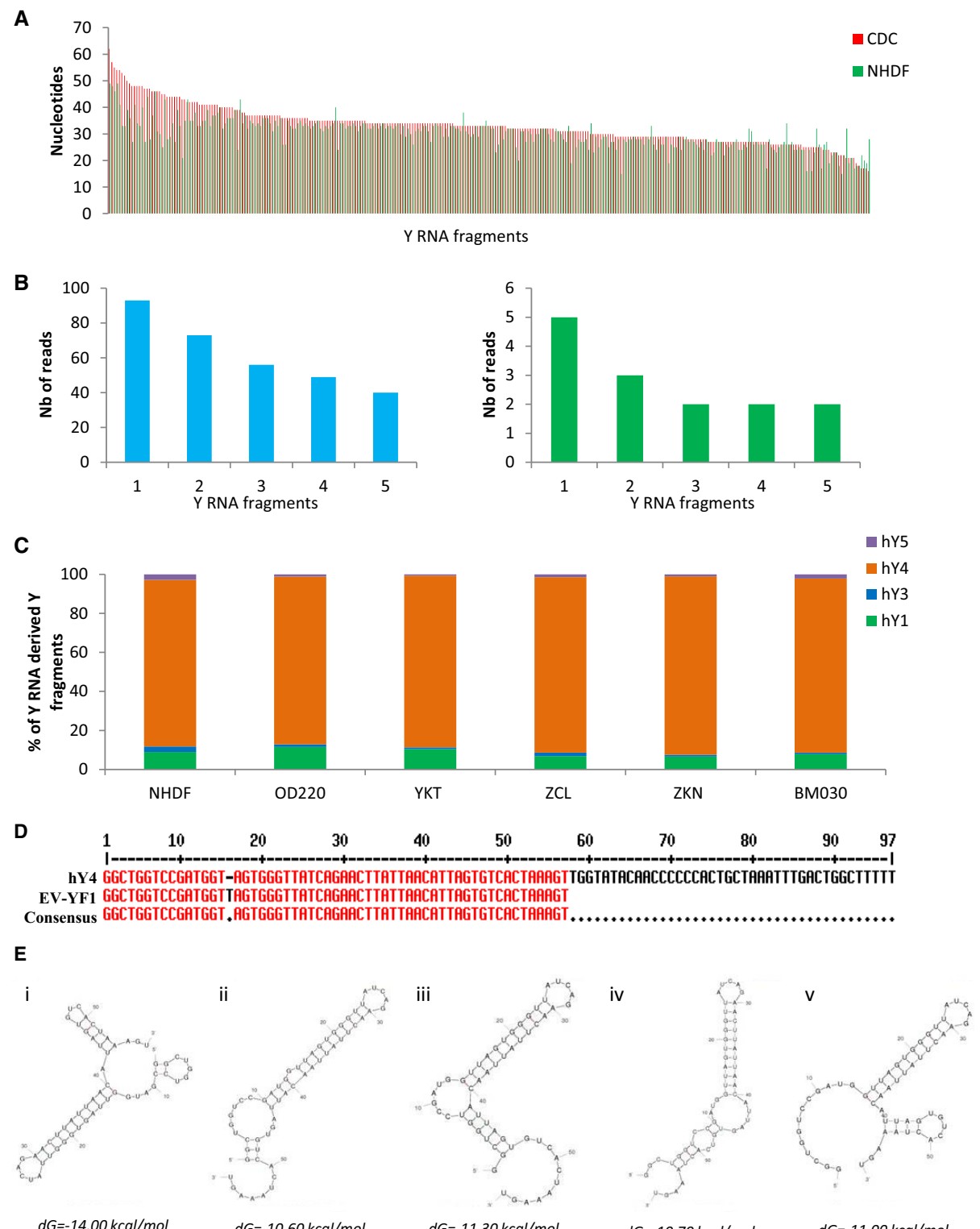

**Figure 3.  CDC-EVs Y RNA fragment length, distribution, and alignment.**

A  Graph representing the nucleic acid length of the 304 common Y RNA fragments between CDC-EVs and NHDF-EVs.

B  Graphical depictions of the abundance of the five most abundant unique Y RNA fragments in CDC-EVs (left) and in NHDF-EVs (right) according to the number (Nb) of reads obtained by RNA-seq.

C  Percentage of Y RNA fragments in CDC-EVs from different CDC donors derived from each full-length Y RNA (hY1, hY3, hY4, hY5).

D  Sequence alignment between hY4 and EV-YF1 reveals a thymine insertion at position 16 in EV-YF1 (score: 99.0 bits, identities 56/57; 98%).

E  Predicted secondary structures of EV-YF1 by UNAFold (*dG*: delta Gibbs free energy).

**Table 3.  Oligoribonucleotide sequences.**

| Ys | 5′-GAUGUUAUUAUCGUAGUAGAUGAAU AAUCGGUGCUACGAUUAUGAGUGUCAG UCGCC-3′ |
|---|---|
| EV-YF1 | 5′-GGCUGGUCCGAUGGUUAGUGGGUUA UCAGAACUUAUUAACAUUAGUGUCA CUAAAGU-3′ |
| EV-YF1-fluo | 5′-/5RhoR-XN/GGCUGGUCCGAUGGUUAG UGGGUUAUCAGAACUUAUUAACAUUAGUGU CACUAAAGU-3′ |

## Elevated EV-YF1 content within CDC-EVs correlates with *in vivo* CDC potency

CDCs have a range of potency depending on the donor (age, gender, comorbidities, etc.) (Mishra *et al*, 2011; Cheng *et al*, 2014a). To test whether the abundance of EV-YF1 within CDC-EVs correlates with *in vivo* functional benefit of the parent CDCs, we utilized an established mouse model of MI (Ibrahim *et al*, 2014). Potent CDC lines (i.e., those which increased post-MI ejection fraction after intramyocardial injection compared to placebo (Appendix Fig S2)) produced EVs with a higher average abundance of EV-YF1 than non-potent CDCs (Fig 2C). While the CDC lines varied considerably in EV-YF1 abundance, the negative control NHDFs yielded EVs with the lowest expression of EV-YF1 (Fig 2D).

## Packaging and EV-mediated transfer of EV-YF1

To assess the transfer of EV-YF1 via CDC-EVs to target bone marrow-derived macrophages (BMDMs), we transfected a fluorescently conjugated EV-YF1 (EV-YF1-fluo) (sequence in Table 3) into CDCs and then isolated CDC-EVs after 5 days in SF culture (Fig 4A). The same experiment was performed using NHDFs as a control (Appendix Fig S3). By immunocytochemistry (ICC), EV-YF1-fluo showed punctate signals within the cytoplasm of CDCs (Fig 4E); by qPCR, both CDCs and CDC-EVs revealed enhanced expression of EV-YF1 (Fig 4B and C). Together, these data demonstrated successful EV-YF1-fluo transfection into CDCs and packaging of EV-YF1-fluo into CDC-EVs (CDC-EVs [EV-YF1-fluo]). When we examined NHDFs and NHDF-EVs, we found that each expressed EV-YF1 (Appendix Fig S3A and B). Interestingly, the amount of EV-YF1 was significantly lower in NHDF-EVs than in CDC-EVs, which suggests that CDCs specifically package EV-YF1 into EVs (Appendix Fig S3B). Next, to determine whether EV-YF1-fluo

could be transferred to target cells via CDC-EVs, we exposed BMDMs to CDC-EVs [EV-YF1-fluo] (Fig 4A). Two hours later, we observed punctate signals within the cytoplasm of BMDM (Fig 4F) and enhanced EV-YF1 expression (Fig 4D); this expression pattern was also observed in BMDMs treated with NHDF-EVs (Appendix Fig S3C). Following exposure to CDC-EVs directly transfected with EV-YF1-fluo, BMDMs took up EV-YF1-fluo (Fig 4G–I); this could also be achieved by direct EV-YF1-fluo transfection (Fig 4J–L). Based on ICC, EV-YF1 did not overlap with the mitochondrial network within CDCs or BMDMs (Fig 4E and F). Although we did not detect EV-YF1-fluo in the nuclei of CDCs or BMDMs, we cannot exclude the possibility that dispersed molecules of EV-YF1-fluo, not forming visible clumps, may be present within the nucleus with a weak fluorescent intensity undetectable by ICC.

## IL-10 expression is induced by EV-YF1

CDCs limit infarct size in rats by polarizing macrophages (Mϕ) toward a distinct, cytoprotective phenotype (de Couto *et al*, 2015). Exposure of BMDMs to CDC-EVs yielded changes in gene expression similar to those described after transwell culture with CDCs (de Couto *et al*, 2015) (Fig 5A). To determine whether EV-YF1 modulates gene expression, we transfected EV-YF1 or a scrambled oligoribonucleotide control [Ys] (sequence in Table 3) into BMDMs. EV-YF1 recapitulated some, but not all, of the effects of CDC-EVs (Fig 5B). Most strikingly, we found that EV-YF1 induced an 18-fold increase in *Il10* gene expression relative to Ys within 18 h of transfection (Fig 5C), an effect sustained for at least 72 h (Appendix Fig S4A). These findings were in contrast to those observed when BMDMs were treated with LPS, where *Il10* gene expression rapidly decreased after 72 h (Appendix Fig S4A). Consistent with the increased *Il10* transcript levels (Fig 5C), the secretion of IL-10 protein was enhanced in EV-YF1-primed (compared to Ys-primed) BMDMs 48 and 72 h post-transduction (Fig 5D). Although EV-YF1 also increased the expression of pro-inflammatory cytokines *Nos2* and *Tnf-α*, the upregulation was weaker than for *Il10* (~sevenfold and twofold, respectively). While LPS also induced secretion of IL-10 in BMDMs (Appendix Fig S4B), *Nos2* increased much less in EV-YF1-primed BMDMs than in M1 Mϕ (LPS treatment) (Fig 5A and B).

## Cardioprotective role of EV-YF1

IL-10 is well recognized as an endogenous cardioprotective cytokine in I/R (Yang *et al*, 2000; Bolger *et al*, 2002; Stumpf *et al*, 2008). To

---

**Figure 4.  Cytoplasmic localization and expression of EV-YF1-fluo.**

A    Schematic of the protocol for EV-YF1-fluo transfection into CDCs (OD220 donor) followed by the collection and treatment of CDC-EVs into BMDMs.

B–D  Expression of EV-YF1 by qPCR in CDCs (B), CDC-EVs (C), and BMDMs (D) described in (A). Results depict the mean ± SEM of *n* = 3.

E, F  Representative images of EV-YF1-fluo-transfected CDCs (E) and BMDMs treated with CDC-EVs (F). Fluorescently conjugated EV-YF1 (red, EV-YF1-fluo), MitoTracker Green FM (green, MitoT); DAPI (blue).

G–L  Schematic of the protocol for BMDMs treated with directly transfected CDC-EVs (G) or transfected with EV-YF1-fluo (J). Immunocytochemical staining reveals punctate, cytoplasmic localization of EV-YF1-fluo (red) in BMDMs following treatment with directly transfected CDC-EVs (H) or transfection with EV-YF1-fluo (K). BMDMs in (H and K) were stained with CD45 (green) and DAPI (blue). (I and L) EV-YF1 expression in BMDMs following treatments described in conditions (G and J), respectively, compared to their Ys (scrambled oligoribonucleotide) control. Results depict the mean ± SEM of *n* = 3.

Data information: (B–D, I, L) Groups were compared using two-tailed, unpaired, Student's *t*-test; (B) **$P$ = 0.0013; (C) **$P$ = 0.0059; (D) **$P$ = 0.0019; (I) **$P$ = 0.002; (L) **$P$ = 0.0034. (E, F, H, K) Scale bars: 10 μm.

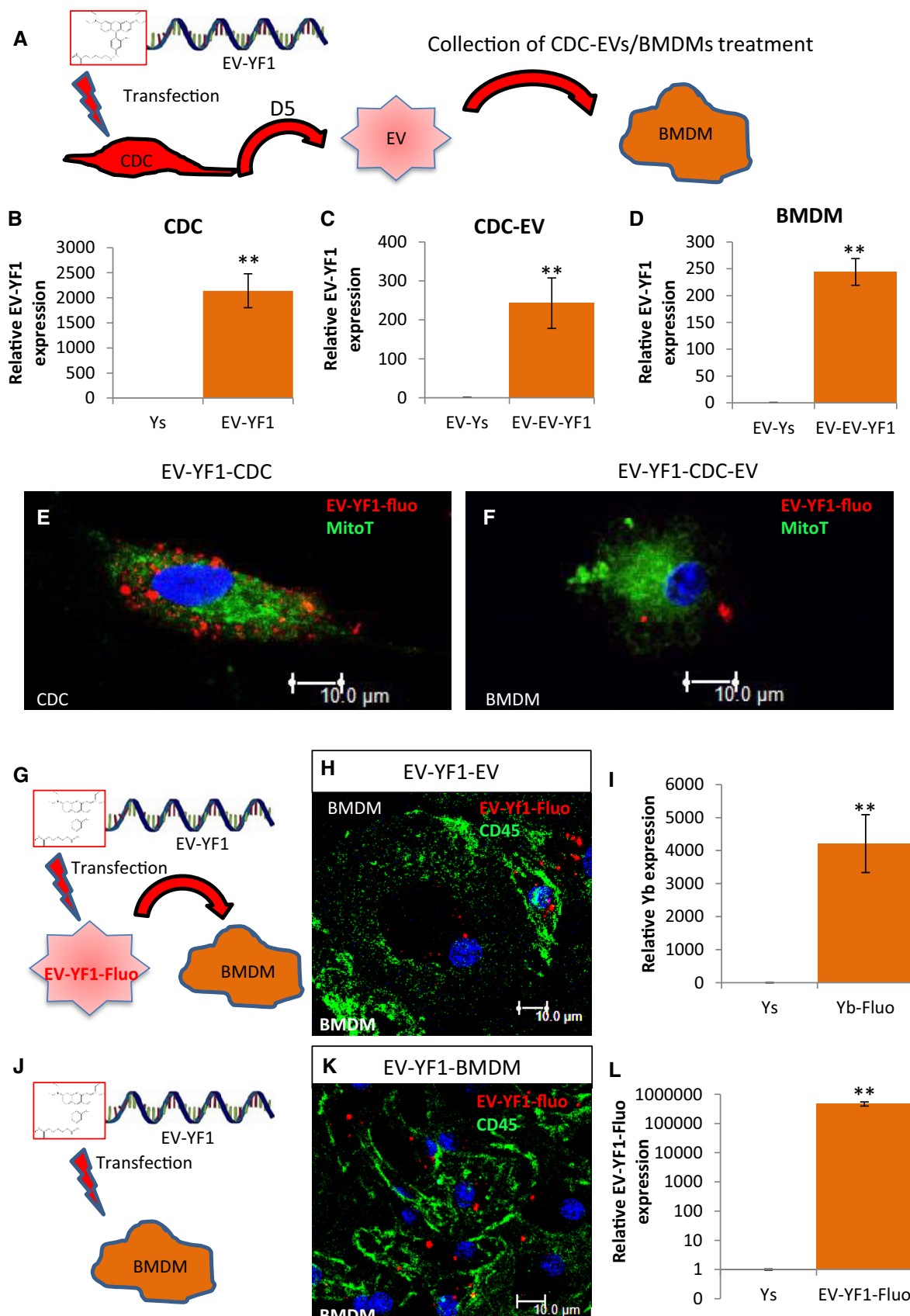

**Figure 4.**

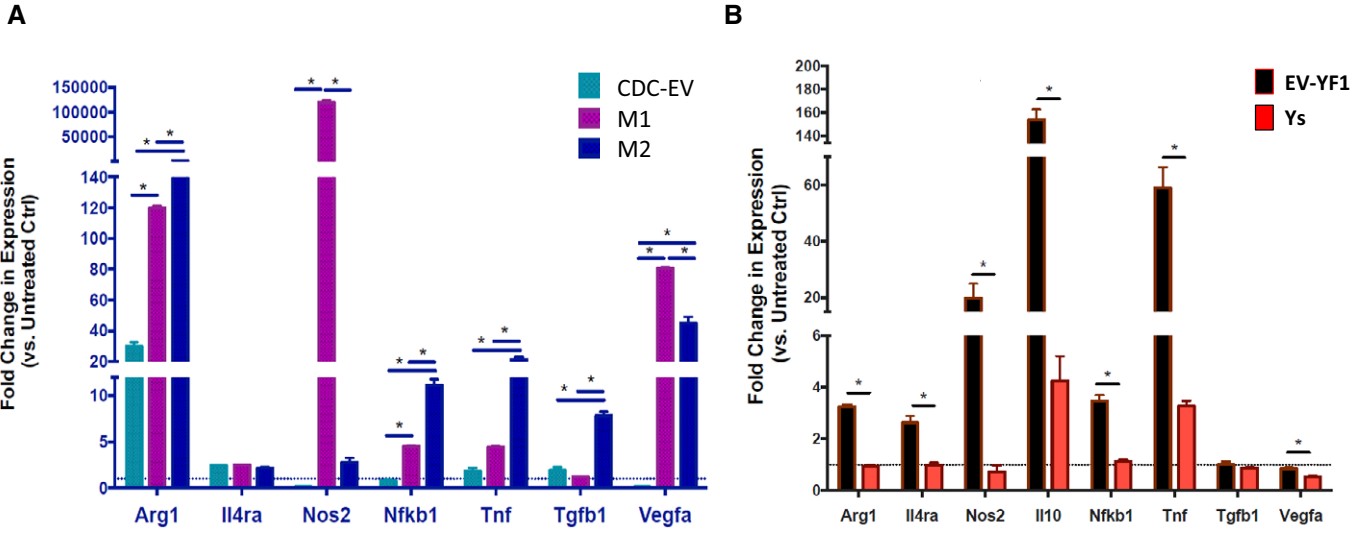

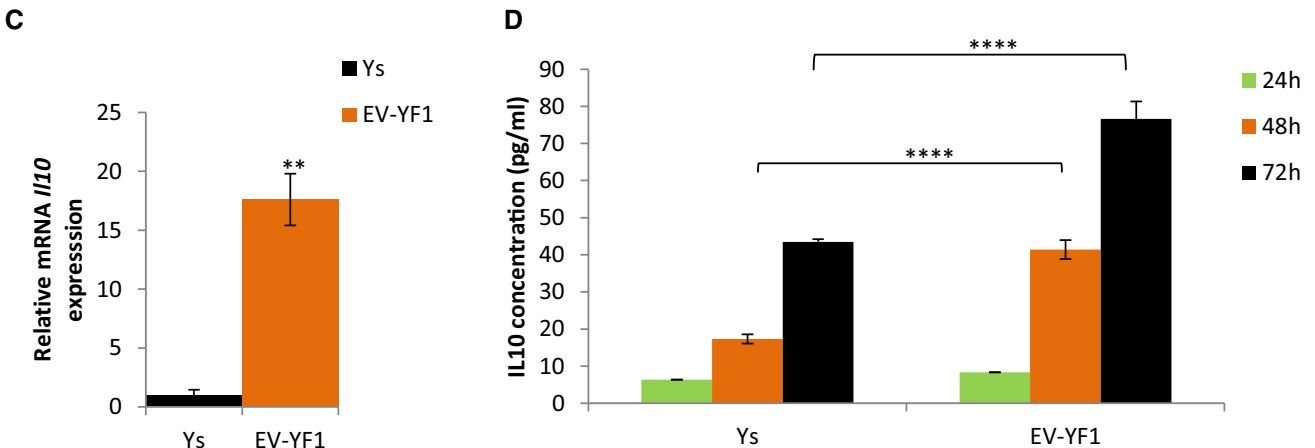

**Figure 5. EV-YF1 modulates IL-10 expression.**

A   Gene expression profile by qPCR of BMDMs polarized toward M1 (IFNγ and LPS), M2 (IL-4 and IL-13) or treated with CDC-EVs (versus untreated control BMDM, dotted line). Results depict the mean ± SEM of two independent experiments, $n = 3$ each. Statistical significance was determined using multiple $t$-tests followed by Holm–Sidak's multiple corrections test; *$P < 0.05$.

B   Gene expression profile by qPCR of BMDMs primed with EV-YF1 or Ys (versus untreated control BMDM, dotted line). Results depict the mean ± SEM of two independent experiments, $n = 3$ each. Statistical significance was determined using multiple $t$-tests followed by Holm–Sidak's multiple corrections test; *$P < 0.05$.

C   Gene expression of *Il10* in BMDMs following transfection with EV-YF1 or Ys, as determined by qPCR. Results depict the mean ± SEM of two independent experiments, $n = 6$ each. Groups were compared using two-tailed, unpaired, Student's $t$-test; **$P < 0.0044$.

D   Protein secretion of IL-10 from BMDMs at 24, 48, and 72 h following transfection with EV-YF1 or Ys, by ELISA. Results depict the mean ± SEM of an experiment representative of two independent experiments, $n = 6$ each. Groups were compared using one-way ANOVA followed by Tukey's multiple comparisons test; ****$P < 0.0001$.

determine the functional consequence(s) of increased IL-10 secretion in EV-YF1-primed BMDMs, we mimicked I/R *in vitro* (de Couto *et al*, 2015). Neonatal rat ventricular myocytes (NRVMs) were stressed with 75 μM $H_2O_2$ for 15 min (simulating an ischemic phase) and then washed with SF media for 20 min (simulating reperfusion), prior to the addition of EV-YF1- or Ys-primed BMDMs in the presence or absence of anti-IL-10 neutralizing antibody (αIL-10). Stressed ($H_2O_2$) and unstressed NRVMs served as

comparators (Fig 6A). NRVM apoptosis was reduced in coculture with EV-YF1-primed BMDMs (TUNEL[+]α-actinin[+]: 24%, versus Ys-primed BMDMs or NRVMs alone: TUNEL[+]α-actinin[+]: ~45%) (Fig 6B and C). The protective effects of EV-YF1-primed BMDMs were strong, as the percentage of apoptosis decreased to a level comparable to that in unstressed NRVMs (TUNEL[+]α-actinin[+]: 20%). The addition of recombinant IL-10 (rIL-10) to stressed NRVMs (without BMDM) mimicked the benefits of coculture with

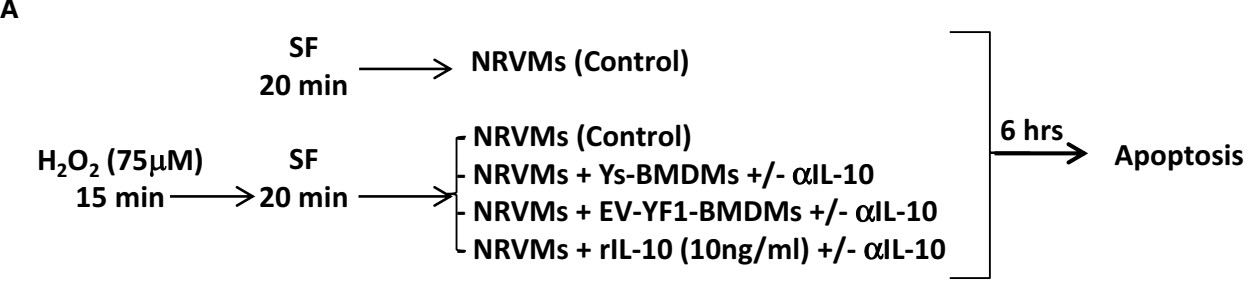

**Figure 6. EV-YF1-primed BMDMs induce IL-10 and protect cardiomyocytes from oxidative stress.**

A Schematic of *in vitro* protocol. NRVMs were cultured with or without 75 μM $H_2O_2$ (15 min), media was replaced with serum-free media (SF) (20 min), and then, Ys- or EV-YF1-primed BMDMs were added in coculture [or recombinant IL-10 (rIL-10, 10 ng/ml) was added]. Six hours later, cells were analyzed for apoptosis. Mean of 2–4 independent experiments in four different wells/experiment.

B Representative images of the cells in (A), stained for TUNEL (green), α-actinin (red), CD45 (white), and DAPI (blue). Scale bars: 10 μm.

C Pooled analyses of TUNEL+ cardiomyocytes (CM). Graphs depict mean ± SEM of 4 replicates. Groups were compared using one-way ANOVA followed by Tukey's multiple comparisons test; †P-values: versus $H_2O_2$ treatment (positive control); *P-values: between treatment groups.

EV-YF1-primed BMDMs (TUNEL+ α-actinin+: 24%) (Fig 6B and C). The protective effects of either EV-YF1-primed BMDMs or rIL-10 were abrogated by αIL-10 neutralizing antibody (Fig 6B and C). To test whether EV-YF1 influences cardiomyocyte survival in

the absence of macrophages, we treated NRVMs with EV-YF1 or Ys and noted a slight, but significant, protective response to oxidative stress (Appendix Fig S5). Taken together, the data support the hypothesis that enhanced secretion of IL-10 from EV-YF1-primed

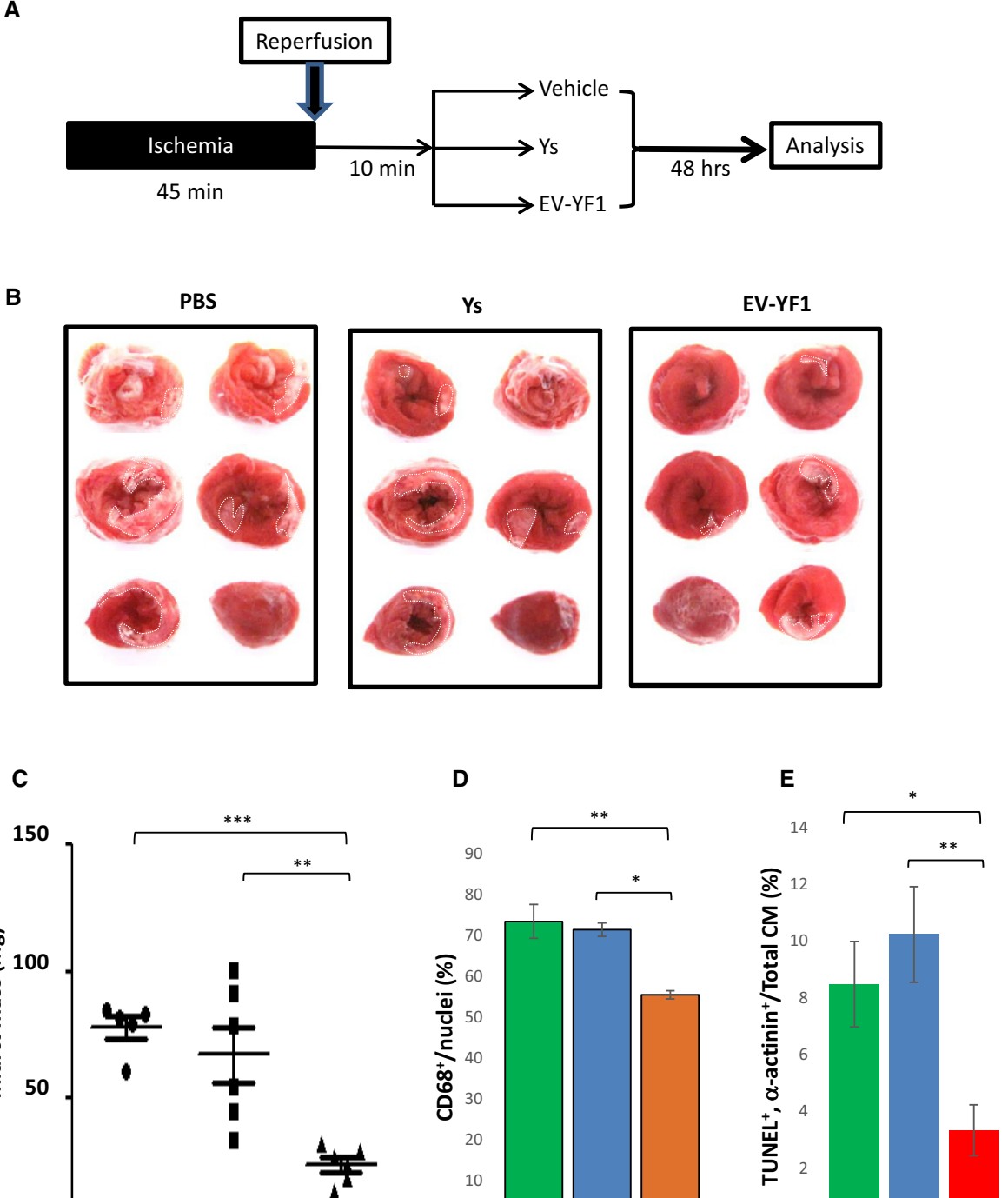

**Figure 7.  EV-YF1 is cardioprotective against I/R injury in rats.**

A  Schematic representation of *in vivo* I/R protocol.

B  Representative TTC-stained hearts from animals at 48 h following I/R injury.

C  Quantitative measurements of TTC-stained hearts, depicted as infarct mass (*n* = 5–6 rats per group). Graphs depict mean ± SEM. Groups were compared using one-way ANOVA followed by Tukey's multiple comparisons test; vehicle versus EV-YF1: ***P = 0.0003; Ys versus EV-YF1: **P = 0.0014.

D  Pooled analysis of CD68[+] cells within the infarct tissue 48 h following I/R injury. Graphs depict mean ± SEM (*n* = 3 rats per group). Groups were compared using one-way ANOVA followed by Tukey's multiple comparisons test; vehicle versus EV-YF1: **P = 0.007; Ys versus EV-YF1: *P = 0.0123.

E  Pooled analysis of TUNEL[+] cardiomyocytes (CM) within the infarct tissue 48 h following I/R injury. Graphs depict mean ± SEM (*n* = 3 rats per group). Groups were compared using one-way ANOVA followed by Tukey's multiple comparisons test; vehicle versus EV-YF1: *P = 0.0377; Ys versus EV-YF1: **P = 0.0075.

BMDMs underlies the cytoprotection of oxidatively stressed cardiomyocytes.

If EV-mediated transfer of EV-YF1 contributes to CDC-mediated cardioprotection (de Couto *et al*, 2015), *in vivo* transfection of EV-YF1 would logically be predicted to mitigate I/R injury. We tested this notion in rats subjected to 45 min of ischemia and 10 min of reperfusion. By random allocation, hearts were then infused with 10 μg of EV-YF1, Ys, or vehicle, with infarct size quantification 2 days later (Fig 7A). Cardiac tissue expression of EV-YF1, assessed 1 h following injection, revealed a 20-fold increase in EV-YF1-treated hearts compared to vehicle controls (Appendix Fig S6A). Animals treated with EV-YF1 exhibited reduced infarct mass (EV-YF1: $24.30 \pm 2.85$ mg, Ys: $67.41 \pm 10.9$ mg, vehicle: $78.33 \pm 4.43$ mg) (Fig 7B and C), a decrease in the number of CD68+ macrophages within the infarct area (EV-YF1: $55.41 \pm 1.01\%$, Ys: $71.33 \pm 1.65\%$, vehicle: $73.35 \pm 4.12\%$; Fig 7D and Appendix Fig S7A), and a decrease in the frequency of apoptotic cardiomyocytes (EV-YF1: $5.08 \pm 1.33\%$, Ys: $16.16 \pm 2.44\%$, vehicle: $13.63 \pm 2.3\%$; Fig 7E and Appendix Fig S7B) compared to animals treated with Ys or vehicle. Additionally, the expression of *Il10* was detectable 24 h later in the hearts of animals treated with EV-YF1; no expression of *Il10* could be detected in the hearts of animals that had been treated with Ys or vehicle (Appendix Fig S6B). Thus, the cytoprotective effects of EV-YF1 seen *in vitro* (Fig 6) are also manifested *in vivo* in a genuine MI model.

## Discussion

Patients who survive a large MI often progress to heart failure (Thune *et al*, 2011; Kikkert *et al*, 2014). To date, CDCs are the only cell type capable of increasing myocardial viability following both acute injury (cardioprotection) (de Couto *et al*, 2015) and established MI (regeneration) (Malliaras *et al*, 2013a,b). CDC-EVs mediate CDC-induced cardioprotection and regeneration (Ibrahim *et al*, 2014). Here, we have (i) discovered that Y RNA fragments comprise the largest small RNA component within CDC-EVs; (ii) identified EV-YF1 as the most abundantly expressed Y RNA fragment; and (iii) assigned unprecedented bioactivity to this fragment, *in vitro* and *in vivo*.

Several RNA-seq analyses of eukaryotic cells have revealed an abundance of small RNA fragments of unknown function. Increasing evidence suggests that some of these RNA fragments, which are likely derived from longer non-coding RNAs (Röther & Meister, 2011), can be functional in both healthy and diseased cells (Hall & Dalmay, 2013). Y RNAs have been implicated in RNA processing and quality control (Stein *et al*, 2005; Fuchs *et al*, 2006; Hogg & Collins, 2007; Wolin *et al*, 2012), as well as DNA replication (Christov *et al*, 2006; Gardiner *et al*, 2009; Krude *et al*, 2009; Kowalski *et al*, 2015), but not in the regulation of gene expression. Y RNA fragments are known to be plentiful in exosomes (Meiri *et al*, 2010; Dhahbi *et al*, 2014; Vojtech *et al*, 2014; Lunavat *et al*, 2015), and most of these fragments appear to derive from hY4 (Vojtech *et al*, 2014; Tosar *et al*, 2015); both observations hold true for EV-YF1 within CDC-EVs. These data support the notion that EV-YF1 is loaded into EVs in a selective manner, rather than as a random by-product of dying cells (Nolte-'t *et al*, 2012; Dhahbi *et al*, 2014).

Y RNAs are well-described binding partners of RNPs, including Ro60 and La, which have nuclear and/or cytoplasmic distribution patterns (Gendron *et al*, 2001). The subcellular localization of Y RNAs, on the other hand, is debated (Pruijn *et al*, 1997; Zhang *et al*, 2011; Hall *et al*, 2013) as to cytoplasmic (O'Brien *et al*, 1993; Peek *et al*, 1993; Gendron *et al*, 2001) or nuclear (Matera *et al*, 1995; Zhang *et al*, 2011). Fluorescently labeled EV-YF1 is visible within the cytosol following transfection, but the effects on transcription of *Il10* presumably occur within the nucleus. These data support previous reports of nuclear-cytosolic shuttling of RNPs and Y RNA and suggest that EV-YF1 may be controlled in a similar manner to modulate transcriptional activation of *Il10*.

Transfection of EV-YF1 induces the expression and secretion of IL-10 in BMDMs, which in turn confers cytoprotection to oxidatively stressed cardiomyocytes *in vitro*. In some animal models of MI, IL-10 shows strong protection against left ventricular (LV) dysfunction but little change in infarct size (Burchfield *et al*, 2008; Stumpf *et al*, 2008). These observations have been attributed to IL-10's ability to inhibit the release of pro-inflammatory cytokines (TNF, IL-6, etc.) (Bolger *et al*, 2002; Asadullah, 2003) and nitric oxide, and to activate matrix metalloproteinases (de Vries, 1995; Mostafa Mtairag *et al*, 2001). However, following myocardial I/R, IL-10 reduces infarct size and improves LV function by suppressing neutrophil infiltration and reducing pro-inflammatory cytokine release (Hayward *et al*, 1997; Yang *et al*, 2000; Stumpf *et al*, 2008; Krishnamurthy *et al*, 2009). Our data demonstrate that EV-YF1, which induces IL-10 expression in Mϕ and suppresses cardiomyocyte death *in vitro*, elicits a similar cardioprotective response *in vivo* by reducing infarct size following I/R. Together, these data support the notion that a focal, concentrated release of IL-10 is required to elicit a cytoprotective anti-apoptotic response (Dhingra *et al*, 2009; Bagchi *et al*, 2013).

IL-10 is a potent anti-inflammatory cytokine and an enticing candidate for the treatment not only of MI, but perhaps also of other inflammatory diseases. Recombinant proteins can be expensive to purify in high concentrations, making them difficult to use in clinical therapeutics. We highlight that EV-YF1, which is actively transferred from CDCs to Mϕ via CDC-EVs, enhances IL-10 expression without the administration of a recombinant protein and does so in a targeted manner via macrophage infiltration. Our findings motivate future testing of EV-YF1 alone, or within EVs, as cell-free off-the-shelf therapeutic candidates to confer cytoprotection.

### Limitations

The present work identifies EV-YF1 as a non-coding RNA enriched within CDC-EVs and establishes EV-YF1's ability to modulate *Il10* transcript levels and enhance IL-10 secretion, but leaves unexplored the mechanism whereby EV-YF1 affects gene expression. The work also fails to establish, beyond the level of plausibility, a role for EV-YF1 as a mediator of the benefits of CDC-EVs during I/R injury. The latter would require selective depletion of EV-YF1 from CDC-EVs, which is not easy to achieve without disrupting the remaining contents. Nevertheless, the demonstration that Y RNA fragments can modulate gene expression motivates the search for

other therapeutically active, non-miR small non-coding RNAs within EVs.

# Materials and Methods

### EVs generation, purification, and transfection

For all experimental procedures, CDC extracellular vesicles (CDC-EVs) were generated from CDCs at passage 4. Normal human dermal fibroblast (NHDF) extracellular vesicles (NHDF-EVs) served as a control (Ibrahim *et al*, 2014).

#### Generation and purification

CDCs and NHDFs were grown to confluence then washed with PBS prior to the addition of serum-free media. Cells were then cultured for 5 days before media collection. The resulting conditioned media was purified with a 0.45-µm filter to remove cellular debris then concentrated with an Amicon 3-kDa centrifugation filter (Millipore). The resulting suspension was utilized for *in vitro* studies. For RNA-seq, this EV suspension was precipitated with ExoQuick (System Biosciences) to isolate exosomal RNA (Appendix Fig S8).

#### Transfection

CDC-EVs were transfected with EV-YF1 or EV-YF1-fluo (5′-linked Rhodamine Red™-X [NHS Ester], IDT) using Exo-Fect (System Biosciences).

### Generation of human cardiosphere-derived cells

Cardiosphere-derived cells (CDCs) were derived as described (Makkar *et al*, 2012). Briefly, heart tissue from six human donors (Table 1), obtained under an IRB-approved protocol, was minced into small pieces and digested with collagenase. Tissue was then plated and cultured on fibronectin (BD Biosciences)-coated dishes, where stromal-like cells and phase-bright round cells grew out spontaneously from the tissue fragments and reached confluence. These cells were then harvested with 0.25% trypsin (GIBCO) and cultured in suspension on ultra-low attachment dishes (Corning) to form cardiospheres. CDCs were obtained by seeding cardiospheres onto fibronectin-coated dishes and passaged. All cultures were maintained at 5% $CO_2$ at 37°C, using IMDM (GIBCO; supplemented with 20% FBS (Hyclone), 1% penicillin/streptomycin, and 0.1 ml 2-mercaptoethanol).

### EV RNA sequencing (RNA-seq)

Sequencing was performed by the Cedars-Sinai Genomics Core (Los Angeles, CA). Library construction was performed according to the manufacturers' protocols using the Ion Total RNA-Seq Kit v2 (Life Technologies). One microgram of total RNA was assessed for quality using the Agilent Bioanalyzer 2100, enriched with magnetic beads, fragmented, ligated with adapters, and reversed-transcribed to make cDNA. The resulting cDNA was barcoded using Ion Xpress™ RNA-Seq Barcode 1-16 Kit and then amplified. RNA-seq libraries were assessed for concentration (Qubit dsDNA HS Assay Kit, Invitrogen) and size (DNA 1000 Kit,

Agilent). Samples were multiplexed and amplified (pooled libraries) onto Ion Sphere™ particles using Ion PI™ Template OT2 200 Kit. Ion Sphere™ particles were then purified and prepared (Ion PI™ Sequencing 200 Kit) for sequencing on an Ion Proton Sequencer. The raw sequencing signal was processed (FASTQ), and the adaptor was trimmed (Torrent Suite software) to obtain 10 million reads per sample.

All reads < 15 nucleotides (nt) after adapter removal were excluded from further analysis. To obtain an integrated view of all types of non-coding RNAs, the filtered reads were aligned to a comprehensive non-coding RNA database (RNACentral Release v1.0) (The RNAcentral Consortium, 2015) downloaded from http://rnacentral.org/, using blast+ toolkit v2.2.30) (Camacho *et al*, 2009) with "blastn-short" mode. An alignment score > 75% (high-scoring segment pair) of the query coverage was used to annotate each read. Reads annotated as "Y RNA" were further aligned to full-length human genomic Y RNA sequences (GenBank Accession Number: NR_004391 (hY1), NR_004392 (hY3), NR_004393 (hY4), and NR_001571 (hY5)).

### NRVM isolation and *in vitro* assay

Neonatal rat ventricular myocytes (NRVMs) were cultured as described (Sekar *et al*, 2009). Briefly, hearts were harvested from 2-day-old Sprague Dawley rats and then ventricles were isolated, minced, and enzymatically digested in a solution of trypsin and collagenase overnight. Cells were resuspended in M199 media (10% FBS, glucose, penicillin, vitamin B12, HEPES, and MEM non-essential amino acids; Gibco) and pre-plated to allow non-cardiomyocyte cell attachment. The resulting NRVM suspension was collected and counted prior to plating for experimental use.

To induce oxidative stress in NRVMs, cells were incubated with 75 µM $H_2O_2$ (Sigma-Aldrich) for 15 min at 37°C prior to media exchange for 20 min, and then, Ys- or EV-YF1-primed BMDMs were added to the NRVM culture dishes. Control NRVMs were treated with or without recombinant rat IL-10 (10 ng/ml) (rIL-10; R&D systems). NRVM-BMDM cocultures and IL-10 treated NRVMs were cultured in the presence or absence of rat IL-10 neutralizing antibody (αIL-10; R&D systems). Cardiomyocyte apoptosis was determined 6 h later with a TdT dUDP Nick-End Labeling (TUNEL, Roche) kit according to the manufacturer's protocol. All samples were co-stained with rabbit α-actinin (Abcam), CD11b (BD Biosciences), and DAPI (Sigma).

### Ischemia/reperfusion rat model

Twelve-week-old female Wistar–Kyoto rats (Charles River Labs) were used for *in vivo* experimental protocols. To induce I/R injury, rats were provided general anesthesia and then a thoracotomy was performed at the fourth intercostal space to expose the heart and left anterior descending coronary artery (LAD). A 7-0 silk suture was then used to ligate the LAD, which was subsequently removed after 45 min to allow for reperfusion. Ten minutes later, 100 µl of EV-YF1, Ys or vehicle was injected into the LV cavity over a period of 20 seconds with aortic cross-clamp. Briefly, 10 µg of EV-YF1 or Ys was incubated in IMDM basal media (Thermo Scientific) with Dharmafect transfection reagent

(Dharmacon) for 10 min at room temperature then resuspended in 100 µl IMDM for injection.

## Histology and immunohistochemistry

### Histology

Two days following I/R injury, 10% KCL was injected into the LV to arrest hearts in diastole. Then, hearts were harvested, washed in PBS, and then cut into 1-mm sections from apex to base, above the infarct zone. Sections were incubated with 1% solution 2,3,5-triphenyl-2H-tetrazolium chloride (TTC) for 20 min in the dark and washed with PBS. Then, sections were imaged and weighed. The infarcted zones (white) were delineated from viable tissue (red) and analyzed (ImageJ software). Infarct mass was calculated according the LV area on both sides of the tissue sections according to the following formula: (infarct area/LV area) × weight (mg).

### Immunohistochemistry

For analyses inflammatory cell distribution, OCT-embedded tissue sections were fixed with 4% PFA and stained with the following primary antibodies for confocal microscopy: rabbit anti-rat α-actinin (Abcam), mouse anti-rat CD68 (Bio-Rad). The appropriate fluorescently conjugated secondary antibodies (Invitrogen) were applied prior to mounting using Fluoroshield with DAPI (Sigma-Aldrich).

## Bone marrow cell isolation and Mϕ differentiation

Femurs were isolated from 7- to 10-week-old Wistar–Kyoto rats. Bone marrow was isolated by flushing with PBS (containing 1% FBS, 2 mM EDTA) then filtering through a 70-µm mesh. Red blood cells were lysed with ACK buffer (Invitrogen) and then resuspended in IMDM (Gibco) containing 10 ng/ml M-CSF (eBioscience) for plating. The media was exchanged every 2–3 days until day 7, at which point bone marrow-derived macrophages (BMDMs) were obtained. BMDMs were transfected with Ys (50 nM) or EV-YF1 (50 nM) using Dharmafect 4 reagent (Dharmacon), treated with LPS (1 µg/ml), or primed toward M1 (100 ng/ml LPS and 50 ng/ml IFN-γ; Sigma-Aldrich and R&D Systems, respectively) or M2 (10 ng/ml IL-4 and IL-13; R&D Systems), the night between days 7 and 8 (~18 h).

## RNA isolation

Cells were washed and collected for RNA isolation using a miRNeasy Mini Kit (QIAGEN) according to the manufacturer's protocol. Exosomal RNA was isolated using the miRNeasy Serum/ Plasma Kit (QIAGEN) according to the manufacturer's protocol. RNA concentration and purity were determined using a NanoDrop Spectrophotometer (Thermo Scientific).

## Quantitative RT–PCR (qPCR)

To assess *Il10* and *EV-YF1* expression, cDNA was synthesized from mRNA using iScript™ cDNA Synthesis Kit (Bio-Rad) according to the manufacturer's protocol. The resulting cDNA was standardized across samples prior to qPCR analysis with iQ™ SEV-YF1R® Green

**Table 4.  qPCR primer sequences.**

| EV-YF1 | 5′-GGCTGGTCCGATG GTTAGTG-3′ | 5′-ACTTTAGTGAC ACTAATGTT-3′ |
|---|---|---|
| HPRT | 5′-GAGAGATCATCTCC ACCAAT-3′ | 5′- ACTTTAGTGA CACTAATGTT-3′ |
| U6 | 5′-GCTTCGGCAGCACAT ATACTAAAAT-3′ | 5′-CGCTTCACGAATTTG CGTGTCAT-3′ |

Supermix (Bio-Rad) on a LightCycler 7900 Fast Real-Time PCR System (Roche Applied Science). Relative gene expression was determined by the ΔΔCt method. Primers were ordered from Integrated DNA Technologies (IDT) (Table 4).

## Enzyme-linked immunosorbent assay (ELISA)

Protein levels of secreted IL-10 were determined using an IL-10 ELISA kit (R&D systems) according to the manufacturer's protocol. Conditioned media collected from Ys- and EV-YF1-primed BMDMs at 24, 48, and 72 h following transfection were utilized to determine secreted levels of IL-10.

## Cellular transfection

To overexpress EV-YF1, Ys, or EV-YF1-fluo, cells (BMDMs or CDCs) were transfected with EV-YF1, Ys, or EV-YF1-fluo at a final concentration of 50 nM using Dharmafect 4 reagent (Dharmacon), according the manufacturer's protocol.

## Statistics

Results are expressed as mean ± SEM. Groups were compared using two-tailed, unpaired, Student's *t*-test, multiple *t*-tests followed by Holm–Sidak's multiple corrections test, or one-way ANOVA followed by Tukey's multiple comparisons test (*$P < 0.05$, **$P < 0.01$, ***$P < 0.001$, ****$P < 0.0001$). Normal distribution of the data was assessed for experiments with more than six replicates using D'Agostino and Pearson omnibus normality test. All analyses were performed using Prism 5 software (GraphPad).

## Study approval

The Institutional Animal Care and Use Committee approved all animal care and related procedures before study commencement.

**Expanded View** for this article is available online.

## Acknowledgements

We thank Jie Tang for assistance with RNA sequencing. This work was supported by NIH R01 HL124074 to E. Marbán and the Board of Governors of the Cedars-Sinai Medical Center and by CIRM DISC1 NH-W0005A-LA to L. Cambier.

## Author contributions

LC established the hypotheses, designed the study, performed experiments, analyzed data, and wrote the paper. GdC assisted with gene expression profile of polarized BMDMs and design of the study and wrote the paper. AI assisted with CDC-EVs sequencing. AKE performed rat surgeries. JV and WL provided

**The paper explained**

**Problem**

Myocardial infarction (MI) produces ischemia and causes death of cardiomyocytes. Novel cardioprotective strategies (i.e., approaches to limit the extent of permanent injury by limiting MI size) remain highly desirable.

**Results**

Cardiosphere-derived cells (CDCs) represent a candidate cell type for regenerative therapy post-MI as well as their secreted extracellular vesicles (CDC-EVs). We identified a highly enriched Y RNA fragment (EV-YF1) in CDC-EVs that is actively transferred from donor cells (CDCs) to target cells via EVs. The abundance of this fragment in CDC-EVs correlates with the potency of CDCs on the reduction of MI size. This fragment, when overexpressed in macrophages, alters IL-10 expression and secretion, IL-10 being the prototype of anti-inflammatory cytokine. Those macrophages overexpressing this Y RNA fragment confer cytoprotection *in vitro* to oxidatively stressed cardiomyocytes when cocultured. *In vivo*, using a rat model of ischemia/reperfusion injury (I/R), EV-YF1, when injected intracoronary few minutes following I/R, confers cardioprotection by decreasing infarct size.

**Impact**

Adult CDC-based therapy constitutes the only treatment that mediates an increase in viable tissue as well as a decrease in scar following I/R injury. CDCs mediate their beneficial effects via secretion of EVs. EVs show certain advantages over CDCs therapy as they are cell-free vesicles with the potential to become an off-the-shelf therapy. Here, we have discovered that CDC-EVs contain high levels of a Y RNA fragment that can alter gene expression in a salutary manner. This Y RNA fragment may itself be useful therapeutically, or its levels might be enhanced in EVs to accentuate their efficacy.

technical assistance. MK, RRS, and LM assisted with CDC potency data. EM established the hypotheses, designed the study, co-analyzed data, and wrote the paper.

## Conflict of interest

EM holds equity in, and serves as unpaid advisor to, Capricor Inc. AI, MK, RRS, and LM are employees of Capricor. The other authors declare that they have no conflict of interest.

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
