## [Review Process File · EMBO Molecular Medicine]

Y RNA fragment in extracellular vesicles confers cardioprotection via modulation of IL-10 expression and secretion

Linda Cambier, Geoffrey de Couto, Ahmed Ibrahim, Antonio K. Echavez, Jackelyn Valle, Weixin Liu, Michelle Kreke, Rachel R. Smith, Linda Marbán, Eduardo Marbán

Corresponding author: Eduardo Marbán, Cedars-Sinai Heart Institute

Review timeline:

Submission date:	05 August 2016
Editorial Decision:	05 October 2016
Revision received:	30 November 2016
Editorial Decision:	16 December 2016
Revision received:	21 December 2016
Accepted:	09 January 2017

Transaction Report:

Editor: Roberto Buccione

1st Editorial Decision

05 October 2016

Thank you for the submission of your manuscript to EMBO Molecular Medicine. We are very sorry that it has taken much longer than usual to get back to you on your manuscript.

In this case, we experienced unusual difficulties in securing three willing and appropriate reviewers. Further to this, one reviewer (#1), despite multiple chasers, failed to deliver his/her report. As a further delay cannot be justified I have decided to proceed based on the two available evaluations.

As you will see, while reviewer 3 is more positive, reviewer 2 is much more reserved and raises fundamental concerns on the appropriateness of the models and methodologies used (a point mentioned also by #3 however), and also feels that the data do not support the main conclusions.

In conclusion, while publication of the paper cannot be considered at this stage, we would be pleased to consider a revised submission, with the understanding that the Reviewers' concerns must be addressed (as explained further below) and that acceptance of the manuscript will entail a second round of review.

After further internal discussion and reviewer cross commenting, we reached a consensus on the way forward. I will not go into too much detail but just clarify some of the raised issues. Specifically, we will not be asking you to experimentally address the following points mentioned by reviewer #2 but simply to provide a clear explanation/justification. All other points by the two reviewers must also be fully addressed, but including with additional experimental data where

appropriate.

Point 1. Given that there were no significant medical comorbidities declared, there is no strong evidence to suggest that the regenerative performance of the CDCs varies between the different cell lines.

Point 3 on Fig. 3C. You could include placebo or dermal fibroblast treated animals as a comparison. We think it would be acceptable to include historical controls rather than a new series of animals because this effect has been well described many times.

Point 6a. We acknowledge that neonatal rat cardiomyocytes are the routine platform used for this type of experiment.

I apologise again for the delay and I look forward to seeing a revised form of your manuscript as soon as possible.

***** Reviewer's comments *****

Referee #2 (Comments on Novelty/Model System):

Human CDC-EVs are tested on adult rat bone marrow cells and on neonatal rat cardiomyocytes. At least it is important to use adult rat bone marrow cells with adult rat cardiomyocytes.

Referee #2 (Remarks):

The authors report that that a Y RNA fragment EV-EF1 inside the CDC-EVs confers cardioprotection by increasing macrophage IL-10 secretion which should protect cardiomyocytes from apoptosis. The study is interesting. However the results are confusing, unprecise and too preliminary to conclude, notably for the in vivo experiments. Furthermore, methodological problems have also to be resolved.

1) Human cardiosphere -derived cells have to be characterized in order to show that the cells obtained with these 6 donors are similar... Indeed, as stated in the text: CDCs have a range of potency depending on the donor ! This could also be due to different origin or maturation of the isolated CDCs. Thus Flow cytometry analysis for the CDCs of all of the 6 donors have to be added to the manuscript to demonstrate that the isolated CDCs are "phenotypically" identical.

2) Figure 1a represents the results obtained using CDC-EVs isolated from a 3-years old child. How is this representative of CDC-EVs isolated from adult hearts ? The figure 2a is not the pooled data from all the 6 donors corresponding to the Figure 1a as stated in the text. Thus please add the pooled data corresponding to the Figure 1a and discuss the relevance of showing as representative data, CDC-EVs isolated from a 3-years old child.

3) Figure 3: is confusing and its legend is not enough precise.

Figure 3A: is this figure the corresponding figure of the figure 1a which should represent the mean of the results of the percentages of the small RNAs in CDC-EVs from different donors ? How many donors ? Which donors, all of the 6 ?

Figure 3C: It would be more interesting to have the individual values for all the donors. Indeed, there is no difference between the donor ZKN and ZCI concerning the EV-YF1 abundance. How can you explain the difference concerning the Ejection Fraction ? How can you explain that 3/6 CDC-EVs worsen the Ejection Fraction ? What is the evolution of the EF in sham animals injected with these CDC-EVs ?

ZCL: ZCI in Figure 3D.

In the text, the message of the Figure 3 should be modulated and the fact that 3 of the 6 CDC-EVs worsen the EF has to be discussed. Furthermore control animals (sham animals) have also to be injected with CDC-EVs.

4) Figure 4C and D: The relative EV-YF-1 expression after Ys transfection (which could be

considered as the "physiological" conditions) has to be evaluated versus EVs from NHDF. I understand that to highlight the mechanism of transfer, it is necessary to study the EV-YF1 transfected CDCs or EVs. However, what is the physiological relevance of this process when only few EV-EF1 fluo is observed in BMDM (see Figure 4F), whereas the relative expression of Ev-EF1 is 1000 x and 200x increased in CDC and CDC-EV ?

Minor point: Mistake of legend of the vertical bar in Figure 4C.

5) Figure 5: The number of different experiments is not indicated.

Figure 5A: the results of the control (untreated BMDMs) have to be added to the figure. The results of the other conditions have to be related to this control. Is nothing statistically significant?

Figure 5B: Is nothing significant ? Why do you focus on IL-10 and not on TNF-alpha ?

Figure 5D: The IL-10 concentration increases with time. What happens after 72h ? Is the IL-10 increase also true in VIVO in rats injected with EV-EF-1 ?

6) Figure 6.

There are several methodology problems in this figure.

a) Why do you test the effect of primed adult BMDMs on neonatal rat cardiomyocytes ? To evaluate physiological relevance you experiments have to be done with adult rat cardiomyocytes, which will react differently to oxidative stress than neonatal cardiomyocytes.

b) To my opinion, an in vitro model to mimic ischemic /reperfusion is to culture the cells in an anoxic environment and then to increase the oxygen percentage. Stimulation with H₂O₂ is not correct to mimic ischemia/reperfusion.

c) Please increase the number of experiments: 2-4 is not sufficient.

d) Figure B is of very poor quality and is not acceptable. No scale bars and the images seem not to be all on the same magnification.

7) Figure 7 presents preliminary results. No evidence that the reduced infarct mass is linked to increase of IL-10 and reduction of cardiomyocyte apoptosis. What is the target of EV-YF1 in this mouse model ? Do you think that 10 min after reperfusion you have a massive infiltration of Bone marrow cells ? or do you have direct effect on cardiomyocytes ? This is not clear and additional experiments have to be performed to understand how this is working in vivo and how the cardiac mass is decreased after EV-YF1 injection compared to control animals.

8) The results of the supplemental figures are not mentioned in the text. Supplemental Figure 3: no number of experiments.

Referee #3 (Remarks):

This report outlines a novel role for extra-vesicle sourced Y RNA fragment as conferring cellular protection when delivered after ischemic reperfusion injury. Although this is a very interesting study that provides plausible evidence for a role in myocardial protection, I have several questions:

1. Overall, there is a lack of appropriate editorial review. Supplementary figures are not sequentially numbered. Figure 2 is mistakenly referred to as showing pooled data from all 6 CDC donors- this is outlined in Figure 3A. Several terms are not defined at first use (e.g., BMDM). Many symbols are not reproduced in the PDF but are represented as squares.

2. As outlined in the text, exosomes are a specific sub-population of small (30-150 nm) extra-cellular vesicles (EVs) suggesting that the EVs studied (Supplementary Figure 2 mean size \approx 150 nm) likely membrane-derived microvesicles rather than smaller intracellularly generated exosomes.

3. EV from commercial normal human dermal fibroblasts (NHDF) are used as a cellular control. It would be more appropriate to use NHDF-EVs sourced from the CDC donor as the changes noted may be attributable to variability in the donors rather than cell type.

4. What cell line was used for the experiments outlined in Figure 5? Given the findings in Figure 3D this needs to be outlined. Changes seen in macrophages derived from bone marrow may not faithfully reproduce responses seen following exposure of cardiac macrophages to EV-YF1 or Ys. Finally, the divergent effects of CDC-EVs or EV-FF1 transduction on Nos2 and Tnf deserve comment.

5. What was the transduction efficiency of EV-YF1? Was the production of EV-YF1 increased and sustained in vivo? What effect EV-YF1 have on the overall myocardial function and were these effects maintained beyond 48 hours?

6. The statistical analysis needs to be expanded. Presumably an analysis of variance was performed. How were multiple comparisons accounted for?

1st Revision - authors' response

30 November 2016

Referee #2 (Comments on Novelty/Model System):

Human CDC-EVs are tested on adult rat bone marrow cells and on neonatal rat cardiomyocytes. At least it is important to use adult rat bone marrow cells with adult rat cardiomyocytes.

REPLY: Adult cardiomyocytes dedifferentiate quickly in primary culture, making co-culture experiments difficult, but we have added data testing the effects of EV-YF1 on adult cardiomyocytes in the post-ischemic heart, finding a reduction in apoptosis (new Fig. 7E).

Referee #2 (Remarks):

The authors report that that a Y RNA fragment EV-EF1 inside the CDC-EVs confers cardioprotection by increasing macrophage IL-10 secretion which should protect cardiomyocytes from apoptosis. The study is interesting. However the results are confusing, unprecise and too preliminary to conclude, notably for the in vivo experiments. Furthermore, methodological problems have also to be resolved.

1) Human cardiosphere -derived cells have to be characterized in order to show that the cells obtained with these 6 donors are similar... Indeed, as stated in the text: CDCs have a range of potency depending on the donor ! This could also be due to different origin or maturation of the isolated CDCs. Thus Flow cytometry analysis for the CDCs of all of the 6 donors have to be added to the manuscript to demonstrate that the isolated CDCs are "phenotypically" identical.

REPLY: All CDCs from different donors were isolated from cardiac tissue and cultured according to our standard protocol, as described previously (Smith, Barile et al., 2007), implying a similar degree of maturation. Each of the donors' CDCs were examined by Flow cytometry to assess cell surface marker expression of CD105, c-Kit, CD31, CD90, CD45 and DDR2 to phenotypically characterize and distinguish our cells from other cell populations. While the CDCs derive from different donors (age, ethnicity, mortality, and sex; **Table 1**), the surface marker expression is consistent (**Table 2**) and conforms to the archetypal expression pattern observed with CDCs (Cheng, Shen et al., 2012).

2) Figure 1a represents the results obtained using CDC-EVs isolated from a 3-years old child. How is this representative of CDC-EVs isolated from adult hearts ? The figure 2a is not the pooled data from all the 6 donors corresponding to the Figure 1a as stated in the text. Thus please add the pooled data corresponding to the Figure 1a and discuss the relevance of showing as representative data, CDC-EVs isolated from a 3-years old child.

REPLY: Although a three-year-old is not considered an adult, the human heart is fully developed by the first year of life. While age undoubtedly influences cardiac tissue, we have not found any significant differences in potency of CDCs derived from young or old donors (*Cf.* EF in MI mouse model, **Appendix Figure S2**). We've shown that CDCs derived from the OD220 donor have a similar surface marker expression profile to all other CDCs (**Table 2**) and that CDCs from this donor are cardioprotective and cardioregenerative (de Couto, Liu et al., 2015, Ibrahim, Cheng et al., 2014). Thus, we've chosen to use the EVs derived from this donor as a representative EV population for our studies.

Figure 2A was mislabeled (we apologize) and had now been corrected. The caption for Fig 3A shows pooled data representing the small RNA content in EVs derived from CDCs from the 6 different donors. The text (page: 4) has also been revised to reflect these changes: *"Fig 3A shows pooled data from 6 different CDC donors with distinct demographic properties (Table 1) but identical surface marker expression (Table 2)"*.

3) Figure 3: is confusing and its legend is not enough precise.

Figure 3A: is this figure the corresponding figure of the figure 1a which should represent the mean of the results of the percentages of the small RNAs in CDC-EVs from different donors ? How many donors ? Which donors, all of the 6 ?

REPLY: We have clarified the details of CDC-EVs (RNA percentage and number of donors) in **Figure 3 Legend** and within the text (page: 4 and 23):

“Fig 1A shows a representative pie chart from one donor (OD220), and Fig 3A shows pooled data from CDCs from the 6 different donors with the demographic properties cited in table 1.”

Figure 3C: It would be more interesting to have the individual values for all the donors.

REPLY: We have highlighted the individual changes in ejection fraction (EF) between CDC donors, as well as a saline-injected control (Placebo), in the new supplemental figure (**Appendix Figure S2**).

Indeed, there is no difference between the donor ZKN and ZCI concerning the EV-YF1 abundance. How can you explain the difference concerning the Ejection Fraction ? How can you explain that 3/6 CDC-EVs worsen the Ejection Fraction ? What is the evolution of the EF in sham animals injected with these CDC-EVs ?

ZCL: ZCI in Figure 3D.

In the text, the message of the Figure 3 should be modulated and the fact that 3 of the 6 CDC-EVs worsen the EF has to be discussed. Furthermore control animals (sham animals) have also to be injected with CDC-EVs.

REPLY: We denote CDC potency based on improvements in EF 21 days post-MI. Thus, ZKN ($\Delta EF\%$: 11.2%) and ZCI ($\Delta EF\%$: -5.8%) were classified accordingly (**Appendix Figure S2**). When we tested the abundance of EV-YF1 in their respective EVs, the expression of EV-YF1 was similar between both donors and resembled the expression pattern observed in potent CDCs (**Fig 3D**). Therefore, we concluded in the text that enrichment of EV-YF1 abundance may not completely account for all of the differences that distinguish CDC potency and emphasized these observations within the text (page 5):

*“While the CDC lines varied considerably in EV-YF1 abundance, the negative control NHDFs yielded EVs with the lowest expression of EV-YF1 (**Fig 3D**).”*

Additionally, the 3 CDC lines that were classified as non-potent had an effect on EF (although to a lesser extent than potent CDCs). In fact, the non-potent CDCs prevented the decline of EF, whereas placebo showed a negative change, following MI (**Fig 3C** and **Appendix Figure S2**).

It is important to note that the parent CDCs were used to assay potency (not secreted EVs), while EV-YF1 is derived from EVs secreted from their respective CDCs. In this context, saline-treated placebo group treatment is considered as a negative control.

The evolution of the EF post-MI of the CDC-injected animals was not monitored beyond 21 days. CDC treatment has been shown to improve EF at this time point, when myocardial scar is well established (Chimenti, Smith et al., 2010, Smith et al., 2007). In other preclinical studies the benefits have been sustained for as long as the animals were observed (e.g., up to 6 months (Malliaras, Li et al., 2012)).

The mislabeling in **Figure 3D** has been corrected from ZCL to ZCI.

4) Figure 4C and D: The relative EV-YF-1 expression after Ys transfection (which could be considered as the "physiological" conditions) has to be evaluated versus EVs from NHDF. I understand that to highlight the mechanism of transfer, it is necessary to study the EV-YF1 transfected CDCs or EVs. However, what is the physiological relevance of this process when only few EV-EF1 fluo is observed in BMDM (see Figure 4F), whereas the relative expression of Ev-EF1 is 1000 x and 200x increased in CDC and CDC-EV ?

Minor point: Mistake of legend of the vertical bar in Figure 4C.

REPLY: We have included a new figure (**Appendix Figure S3**), which highlights the expression of EV-YF1 in NHDFs (**Appendix Figure S3A**), their secreted EVs (**Appendix Figure S3B**) and BMDMs treated with those EVs (**Appendix Figure S3C**) following transfection of NHDFs with Ys or EV-YF1. Although the immunofluorescence method highlights the expression of fluorescent EV-YF1 oligonucleotide, it does not provide a quantitative approach. Thus, we've chosen to assess the expression of EV-YF1 by qPCR, which allowed us to determine quantitatively the active transfer of EV-YF1 from CDCs to EVs, and then uptake from EVs to their target cells (BMDMs) sufficient to induce an effect (**Figure 4, A-D**).

The mistake in Figure 4C has been corrected.

5) Figure 5: The number of different experiments is not indicated.

REPLY: The number of experiments has been added to each of the figure legends.

Figure 5A: the results of the control (untreated BMDMs) have to be added to the figure. The results of the other conditions have to be related to this control. Is nothing statistically significant?

REPLY: We have revised **Fig 5A**, and the figure legend, to reflect statistical differences (*) between treatment groups. Each treatment group reflects the fold change versus untreated control BMDM (dotted line).

Figure 5B: Is nothing significant ?

REPLY: As above, we have revised **Fig 5B**, and the figure legend, to reflect statistical differences (*) between treatment groups. Each treatment group reflects the fold change versus untreated control BMDM (dotted line).

Why do you focus on IL-10 and not on TNF-alpha ?

REPLY: Although several genes are affected by EV-YF1 (**Fig 5B**), we focused on *Il10* gene expression over *Tnf* since both CDCexo and EV-YF1 induced *Il10* to a greater extent than *Tnf* (**Fig 5, A-B**). While both *Il10* and *Tnf* were induced following EV-YF1 treatment, the ratio of these genes suggested that BMDMs treated with EV-YF1 would lead to an anti-inflammatory and cytoprotective response. This was validated *in vitro*, where EV-YF1-primed BMDM protected cardiomyocytes from oxidative stress via enhanced secretion of IL-10 (**Fig 6**).

Figure 5D: The IL-10 concentration increases with time. What happens after 72h ? Is the IL-10 increase also true in VIVO in rats injected with EV-EF-1 ?

REPLY: BMDMs were transfected 7-10 days after BM isolation. We did not extend our expression analysis of *Il10* beyond 72h since we hypothesized that the cytoprotective/cardioprotective effects of EV-YF1 were acute (<72hrs) (de Couto et al., 2015). This hypothesis was subsequently validated *in vitro* (**Fig 5**) and *in vivo* (**Fig 6**). To further support these findings, we examined *Il10* expression within the heart 24h after I/R in rats that had been treated with EV-YF1, Ys control, or vehicle. While no *Il10* expression was detected in animals treated with the scrambled control (Ys) or vehicle, EV-YF1 was detected in 4/6 animals treated (**Appendix Figure S6B**).

6) Figure 6. There are several methodology problems in this figure.

a) Why do you test the effect of primed adult BMDMs on neonatal rat cardiomyocytes ? To evaluate physiological relevance you experiments have to be done with adult rat cardiomyocytes, which will react differently to oxidative stress than neonatal cardiomyocytes.

REPLY: The yield and viability of adult rat cardiomyocytes is extremely low for effective *in vitro* analyses. Thus, we have chosen to use neonatal rat cardiomyocytes, which is a commonly used cell type to assess oxidative stress (Chlopikova, Psotova et al., 2001, de Couto et al., 2015, Ibrahim et al., 2014). We have also added data testing the effects of EV-YF1 on adult cardiomyocytes in the post-ischemic heart, finding a reduction in apoptosis (new Fig. 7E).

b) To my opinion, an in vitro model to mimic ischemic /reperfusion is to culture the cells in an anoxic environment and then to increase the oxygen percentage. Stimulation with H₂O₂ is not correct to mimic ischemia/reperfusion.

REPLY: We agree that culturing NRVMs in an anoxic environment and then increasing the oxygen percentage is a good model to mimic ischemia/reperfusion. However, this experimental setup is prone to error since it is challenging to maintain low O₂ levels over an extended period of time. Therefore, we have chosen the published and reproducible method of H₂O₂ stimulation to induce oxidative stress in NRVMs (de Couto et al., 2015, Ibrahim et al., 2014). This method simulates ischemia-reperfusion (I/R)-induced myocardial injury through increased sodium exchanger (NHE-1) activity, upregulated reactive oxygen species (ROS) production, and accumulation of intracellular Ca²⁺ (Finkel, 2011, Green & Kroemer, 2004, Li, Yan et al., 2012, Rothstein, Byron et al., 2002). Recognizing the limitations of any *in vitro* model, we have additionally provided data in a genuine *in vivo* model of MI (Fig. 7).

c) Please increase the number of experiments: 2-4 is not sufficient.

REPLY: We performed these experiments using NRVMs derived from a pool of 20-30 rat pup hearts. Each experiment (except α -IL-10) was repeated 3 times (total of 4 independent experiments) with an n=4 per experiment; the experiments using α -IL-10 were repeated once (total of 2 independent experiments) with n=4 per experiment. Since we observed low variability between our 4 experimental results (Figure below), we are confident that the replicates are sufficient to demonstrate the obtained results.

d) Figure B is of very poor quality and is not acceptable. No scale bars and the images seem not to be all on the same magnification.

REPLY: **Figure 6B** has been revised to include higher quality images with scale bars.

7) Figure 7 presents preliminary results. No evidence that the reduced infarct mass is linked to increase of IL-10 and reduction of cardiomyocyte apoptosis.

REPLY: To clarify the role of EV-YF1 in reducing infarct mass, attenuating apoptosis, and increasing IL-10 within the heart we incorporated additional new experiments (**Fig 7, D-E, Appendix Figure S6B, Appendix Figure S7**). We've highlighted that *Il10* expression is detectable within the heart 24h after I/R in 4/6 animals treated with EV-YF1 (versus Ys and PBS; **Appendix Figure S6B**). Furthermore, we've shown a reduction in inflammatory infiltrating CD68+

macrophages (**Fig 7D** and **Appendix Figure S7A**) and TUNEL+ cardiomyocytes (**Fig 7E** and **Appendix Figure S7B**) in EV-YF1-treated animals versus Ys or vehicle (PBS). Together, these data, along with our *in vitro* macrophage data (**Fig 5, C-D; Figure 6**), support our conclusions that EV-YF1, by way of macrophage-induced IL-10 production, reduces infarct mass (**Fig 7B**), inflammation and cardiomyocyte apoptosis.

What is the target of EV-YF1 in this mouse model ?

REPLY: The data presented here highlight the role of EV-YF1 in modulating macrophage *Il10* gene expression. These findings merit further investigation into the mechanism to fully understand how EV-YF1 modulates *Il10* gene expression in macrophages.

Do you think that 10 min after reperfusion you have a massive infiltration of Bone marrow cells ? or do you have direct effect on cardiomyocytes ? This is not clear and additional experiments have to be performed to understand how this is working in vivo and how the cardiac mass is decreased after EV-YF1 injection compared to control animals.

REPLY: Our lab has previously shown that rats treated with CDCs following 20 min of reperfusion, exhibit a reduced infarct size at 48 hrs. The mechanism for this cardioprotective response was found to be related to a distinct polarization shift in the macrophages within the heart (de Couto et al., 2015). While cells, and not EVs, were used in that previous experiment, we've provided significant *in vitro* data demonstrating a specific effect of EV-YF1 on bone marrow-derived macrophages (**Fig 4-6**). Although fewer bone marrow-derived macrophages are found within the ischemic area after 10 mins of reperfusion, in contrast to the influx observed at 24-48 hrs, macrophages are present (monocyte-derived and resident cardiac) although in limited numbers. These macrophages, which become polarized in the presence of EV-YF1, have potent anti-apoptotic effects on surrounding cardiomyocytes (**Fig 6**).

In addition to the effects on macrophages, we have provided new *in vitro* data to examine the effects of EV-YF1 on neonatal rat ventricular myocytes (NRVMs). Here, we demonstrate that overexpression of EV-YF1, in contrast to Ys control, protects NRVMs against oxidative stress (**Appendix Figure S5**). Together, these data demonstrate that EV-YF1 acts on both on cardiomyocytes and macrophages to protect against ischemic injury.

8) The results of the supplemental figures are not mentioned in the text. Supplemental Figure 3: no number of experiments.

REPLY: The results of the supplemental figures were highlighted within the text and are noted below:

Page 4: Exosome-enriched EVs from 6 human CDC donors exhibited typical particle numbers and size distributions compared to normal human dermal fibroblast (NHDF) EVs (NHDF-EVs), as exemplified in Appendix Figure S1A and B.

Page 5: Potent CDC lines (i.e., those which increased post-MI ejection fraction after intramyocardial injection compared to placebo (Appendix Figure S2)) produced EVs with a higher average abundance of EV-YF1 than non-potent CDCs (Fig 3C).

Page 5: The same experience was performed using NHDFs as a control (Appendix Figure S3).

Page 6: NHDFs and NHDF-EVs revealed also enhanced expression of EV-YF1, although the expression in NHDF-EVs is lower than in CDC-EVs, suggesting a specific packaging of EV-YF1 into EVs operates in CDCs (Appendix Figure S3A and B).

Page 6: Two hours later, we observed punctate signals within the cytoplasm of BMDM (Fig 4F) and enhanced EV-YF1 expression (Fig 4D), expression also observed in BMDMs treated with NHDF-EVs (Appendix Figure S3C).

*Page 6-7: Most strikingly, we found that EV-YF1 induced an 18-fold increase in *Il10* gene expression relative to Ys within 18 hrs of transfection (Fig 5C), an effect sustained for at least 72 hrs (Appendix Figure S4A). These findings were in contrast to those observed when BMDMs were treated with LPS, where *Il10* gene expression rapidly decreased after 72 hrs (Appendix Figure*

S4A). While LPS also induced secretion of IL-10 in BMDMs (**Appendix Figure S4B**), *Nos2* increased much less in EV-YF1-primed BMDMs than in M1 M ϕ (LPS-treatment) (**Fig 5A and B**).

Page 7: Taken together, the data support the hypothesis that enhanced secretion of IL-10 from EV-YF1-primed BMDMs underlies the cytoprotection of oxidatively-stressed cardiomyocytes, while a minor direct cytoprotective effect was observed in oxidatively-stressed NRVMs overexpressing EV-YF1 (**Appendix Figure S5**).

Page 7: Expression of EV-YF1 was assessed in heart one hour after injection and showed 20-fold increase compare to vehicle injected hearts (**Appendix Figure S6A**).

Page 8: Animals treated with EV-YF1 exhibited reduced infarct mass compared to animals treated with Ys or vehicle (EV-YF1: 24.30 ± 2.85 mg, Ys: 67.41 ± 10.9 mg, vehicle: 78.33 ± 4.43 mg) (**Fig 7B and C**) as well as a decrease in the number of inflammatory macrophages CD68+ infiltration in the infarct area (**Fig 7D and Appendix Figure S7A**) and a decrease in apoptotic cells (**Fig 7E and Appendix Figure S7B**).

In addition, *Ili10* expression was detected in heart 24h after treatment with EV-YF1 while no *Ili10* expression was detected in heart treated with Ys or vehicle (**Appendix Figure S6B**).

Page 11: For RNA-seq, this exosome suspension was precipitated with ExoQuick (System Biosciences) to isolate exosomal RNA (**Appendix Figure S8**).

We have revised the text for the figure legend of **Appendix Figure S3** to include the number of experiments.

Referee #3 (Remarks):

This report outlines a novel role for extra-vesicle sourced Y RNA fragment as conferring cellular protection when delivered after ischemic reperfusion injury. Although this is a very interesting study that provides plausible evidence for a role in myocardial protection, I have several questions:

1. Overall, there is a lack of appropriate editorial review. Supplementary figures are not sequentially numbered. Figure 2 is mistakenly referred to as showing pooled data from all 6 CDC donors- this is outlined in Figure 3A. Several terms are not defined at first use (e.g., BMDM). Many symbols are not reproduced in the PDF but are represented as squares.

REPLY: All of the **Supplementary Figures** have been cross-checked for sequential numbering. Additionally, **Figure 2A** has been corrected within the text and now mentions 1 donor rather than a pooled dataset (**Fig 3A**).

Page 4: **Fig 1A** shows a representative pie chart from one donor (OD220), and **Fig 3A** shows pooled data from 6 different CDC donors with distinct demographic properties (**Table 1**) but identical surface markers (**Table 2**).

REPLY: The terms were defined at first use (e.g., BMDM, LV), however we noted that many symbols were not properly converted in the PDF (represented as squares). These technical issues have been resolved in this revised manuscript.

2. As outlined in the text, exosomes are a specific sub-population of small (30-150 nm) extra-cellular vesicles (EVs) suggesting that the EVs studied (Supplementary Figure 1 mean size \approx 150 nm) likely membrane-derived microvesicles rather than smaller intracellularly generated exosomes.

REPLY: We've paid particular attention in the text (e.g., Introduction; page 3, paragraph 2) to denote the bioactive CDC-derived particles as "EVs".

3. EV from commercial normal human dermal fibroblasts (NHDF) are used as a cellular control. It would be more appropriate to use NHDF-EVs sourced from the CDC donor as the changes noted may be attributable to variability in the donors rather than cell type.

REPLY: Unfortunately, we are unable to isolate NHDFs from the same CDC donor since our IRB protocol covered only the collection of heart tissue for CDC generation. Furthermore, these tissues were harvested from patients that are deceased and thus impossible for us to collect retroactively. We feel that NHDFs, which are routinely used by our group and others (Chimenti et al., 2010, Ibrahim et al., 2014, Latham, Ye et al., 2013), are an appropriate therapeutically-inert controls.

4. What cell line was used for the experiments outlined in Figure 5? Given the findings in Figure 3D this needs to be outlined.

REPLY: The CDC line used in Figure 5 was OD220; this has been clarified in the legend of **Figure 5**.

Changes seen in macrophages derived from bone marrow may not faithfully reproduce responses seen following exposure of cardiac macrophages to EV-YF1 or Ys.

REPLY: Our lab has previously shown that CDC-mediated macrophage polarization occurs in a similar way between cardiac macrophages, peritoneal macrophages, and BMDMs (de Couto et al., 2015). Since very few cardiac macrophages are isolated from heart tissue, even following ischemic injury, we sought to investigate the effect of EV-YF1 on a readily available source of macrophages (BMDM).

Finally, the divergent effects of CDC-EVs or EV-FF1 transduction on *Nos2* and *Tnf* deserve comment.

REPLY: We focused on *Il10* rather than the pro-inflammatory cytokines *Tnf* and *Nos2* since their expression levels were induced to a lesser extent by EV-YF1 than *Il10* (2- and 7-fold, respectively). Moreover, the gene expression profile induced by EV-YF1, which supports a stronger anti-inflammatory response, is supported by our in vitro co-culture data (**Fig 6**). Nevertheless, we have included comment on both *Nos2* and *Tnf* within the section: "Results- IL-10 expression is induced by EV-YF1" (Pages 6-7).

5. What was the transduction efficiency of EV-YF1? Was the production of EV-YF1 increased and sustained in vivo? What effect EV-YF1 have on the overall myocardial function and were these effects maintained beyond 48 hours?

REPLY: We did not assess transduction efficiency *in vivo*, but when we injected EV-YF1 within the heart following 10 min of reperfusion (Figure 7A), we observed a 20-fold increase in EV-YF1 expression within the heart one hour later (versus vehicle controls; **Appendix Figure S6**). We did not assess EV-YF1 expression beyond that time as we expect uptake and clearance by circulating macrophages.

The effect of EV-YF1 on overall myocardial function was not assessed following I/R. Furthermore, based on previous data, effects of CDC treatment persist for 2 weeks (de Couto et al., 2015)). The focus of this paper was to determine whether EV-YF1 can support acute cardioprotection, as defined by a reduction in infarct size. Thus, while we do acknowledge that cardiac function provides additional supportive information, we believe it is beyond the scope of the paper.

6. The statistical analysis needs to be expanded. Presumably an analysis of variance was performed. How were multiple comparisons accounted for?

REPLY: For analysis of experiments involving only 2 groups, groups were compared using 2-tailed, unpaired, Student's t test.

For analysis of experiments involving more than 2 groups, groups were compared using 1-way ANOVA followed by Tukey's multiple comparisons test or multiple t-tests followed by Holm-Sidak's multiple corrections test.

The P values significance were assigned according: * $p < 0.05$, ** $p < 0.01$, *** $p < 0.001$, **** $p < 0.0001$. All analyses were performed using Prism 5 software (GraphPad).

A detailed explanation of statistical analyses was added in the methods section.

Additional references

- Cheng K, Shen D, Smith J, Galang G, Sun B, Zhang J, Marban E (2012) Transplantation of platelet gel spiked with cardiosphere-derived cells boosts structural and functional benefits relative to gel transplantation alone in rats with myocardial infarction. *Biomaterials* 33: 2872-9
- Chimenti I, Smith RR, Li T-S, Gerstenblith G, Messina E, Giacomello A, Marbán E (2010) Relative Roles of Direct Regeneration Versus Paracrine Effects of Human Cardiosphere-Derived Cells Transplanted Into Infarcted Mice. *Circulation Research* 106: 971-980
- Chlopčiková S, Psotová J, Miketová P (2001) Neonatal rat cardiomyocytes--a model for the study of morphological, biochemical and electrophysiological characteristics of the heart. *Biomed Pap Med Fac Univ Palacky Olomouc Czech Repub* 145: 49-55
- de Couto G, Liu W, Tseliou E, Sun B, Makkar N, Kanazawa H, Arditi M, Marban E (2015) Macrophages mediate cardioprotective cellular postconditioning in acute myocardial infarction. *J Clin Invest* 125: 3147-62
- Finkel T (2011) Signal transduction by reactive oxygen species. *The Journal of Cell Biology* 194: 7-15
- Green DR, Kroemer G (2004) The Pathophysiology of Mitochondrial Cell Death. *Science* 305: 626-629
- Ibrahim Ahmed G-E, Cheng K, Marbán E (2014) Exosomes as Critical Agents of Cardiac Regeneration Triggered by Cell Therapy. *Stem Cell Reports* 2: 606-619
- Latham N, Ye B, Jackson R, Lam B-K, Kuraitis D, Ruel M, Suuronen EJ, Stewart DJ, Davis DR (2013) Human Blood and Cardiac Stem Cells Synergize to Enhance Cardiac Repair When Cotransplanted Into Ischemic Myocardium. *Circulation* 128: S105-S112
- Li R, Yan G, Li Q, Sun H, Hu Y, Sun J, Xu B (2012) MicroRNA-145 protects cardiomyocytes against hydrogen peroxide (H₂O₂)-induced apoptosis through targeting the mitochondria apoptotic pathway. *PLoS One* 7: e44907
- Malliaras K, Li TS, Luthringer D, Terrovitis J, Cheng K, Chakravarty T, Galang G, Zhang Y, Schoenhoff F, Van Eyk J, Marban L, Marban E (2012) Safety and efficacy of allogeneic cell therapy in infarcted rats transplanted with mismatched cardiosphere-derived cells. *Circulation* 125: 100-112
- Rothstein EC, Byron KL, Reed RE, Fliegel L, Lucchesi PA (2002) H₂O₂-induced Ca²⁺ overload in NRVM involves ERK1/2 MAP kinases: role for an NHE-1-dependent pathway. *American Journal of Physiology - Heart and Circulatory Physiology* 283: H598-H605
- Smith RR, Barile L, Cho HC, Leppo MK, Hare JM, Messina E, Giacomello A, Abraham MR, Marban E (2007) Regenerative potential of cardiosphere-derived cells expanded from percutaneous endomyocardial biopsy specimens. *Circulation* 115: 896-908

2nd Editorial Decision

16 December 2016

- 1) Both the scale bars and the corresponding labeling in Fig. 4 are difficult to read. Please provide an improved figure.
- 2) As requested by the reviewer, please indicate the scale bar values in the legends to Figure 6 and Supplement Figure 7.
- 3) The description of panel E is missing in the legend to Figure 7
- 4) The supplemental figures should be combined with the legends in a single PDF with a TOC. Also, the current supplementary figures have the wrong label (Supp. Fig 1, etc). This must be corrected together with the corresponding callouts if necessary in the manuscript (please refer to our author guidelines; <http://embomolmed.embopress.org/authorguide#datapresentationformat>).
- 5) Please correct the "Online Table 1: reference on page 11 to "Table 1".
- 6) We could not find callouts for Table 4 or Appendix Figure 8 in the manuscript
- 7) Please provide the Tables as separate files.

8) As per our Author Guidelines, the description of all reported data that includes statistical testing must state the name of the statistical test used to generate error bars and P values, the number (n) of independent experiments underlying each data point (not replicate measures of one sample), and the actual P value for each test (not merely 'significant' or 'P < 0.05').

I look forward to seeing a revised form of your manuscript as soon as possible.

***** Reviewer's comments *****

Referee #2 (Remarks):

Thank you for improving the quality of the manuscript.

Minor modifications:

Please indicate the corresponding measurements of the scale bars on the legends of Figure 6 and Supplement Figure 7.

Referee #3 (Comments on Novelty/Model System):

This is a very good study that will have a significant impact on the field. Thanks for the chance to review this manuscript.

I also like to express my appreciation for the thorough and thoughtful review which this article has undergone- it speaks well to the thoroughness and quality of editorial review at EMBO Molecular Medicine.

Referee #3 (Remarks):

My concerns have been adequately addressed.

Two of the references were duplicates (Stumpf et al., 2008 and Valadi et al., 2007).

2nd Revision - authors' response

21 December 2016

Response to comments

1) Both the scale bars and the corresponding labeling in Fig. 4 are difficult to read. Please provide an improved figure.

REPLY: The scale bars and corresponding labeling in Fig. 4 were modified for clarity (page 24).

2) As requested by the reviewer, please indicate the scale bar values in the legends to Figure 6 and Supplement Figure 7.

REPLY: We have included the scale bar values to the legends of Figure 6 and Supplement Figure 7.

3) The description of panel E is missing in the legend to Figure 7

REPLY: We've included the description in Figure 7, panel E.

4) The supplemental figures should be combined with the legends in a single PDF with a TOC. Also, the current supplementary figures have the wrong label (Supp. Fig 1, etc). This must be corrected together with the corresponding callouts if necessary in the manuscript (please refer to our author guidelines; <http://embomolmed.embopress.org/authorguide#datapresentationformat>).

REPLY: The supplemental figures, with legends and correct labeling (**Appendix Figure S**), have been combined into a single PDF.

5) Please correct the "Online Table 1: reference on page 11 to "Table 1".

REPLY: We've revised the "Online Table 1: reference on page 11 to "**Table 1**".

6) We could not find callouts for Table 4 or Appendix Figure 8 in the manuscript

REPLY: The reference to **Table 4** has been incorporated into the main text (pages 5 and 6) and **Appendix Figure S8** has been described in the Materials and Methods section (page 11).

7) Please provide the Tables as separate files.

REPLY: We have provided the Tables as separate files.

8) As per our Author Guidelines, the description of all reported data that includes statistical testing must state the name of the statistical test used to generate error bars and P values, the number (n) of independent experiments underlying each data point (not replicate measures of one sample), and the actual P value for each test (not merely 'significant' or ' $P < 0.05$ ').

REPLY: We have revised the legends and figures to include the name of the statistical test, the number of independent experiments (n), and the P value for each test.

Corresponding Author Name: Eduardo Marban
Journal Submitted to: EMBO Molecular Medicine
Manuscript Number: EMM-2016-06924